# HuggingGPT: Solving AI Tasks with ChatGPT and its Friends in Hugging Face

**Yongliang Shen**[1,2,*], **Kaitao Song**[2,*,†], **Xu Tan**[2],
**Dongsheng Li**[2], **Weiming Lu**[1,†], **Yueting Zhuang**[1,†]
Zhejiang University[1], Microsoft Research Asia[2]
{syl, luwm, yzhuang}@zju.edu.cn, {kaitaosong, xuta, dongsli}@microsoft.com

https://github.com/microsoft/JARVIS

## Abstract

Solving complicated AI tasks with different domains and modalities is a key step toward artificial general intelligence. While there are numerous AI models available for various domains and modalities, they cannot handle complicated AI tasks autonomously. Considering large language models (LLMs) have exhibited exceptional abilities in language understanding, generation, interaction, and reasoning, we advocate that LLMs could act as a controller to manage existing AI models to solve complicated AI tasks, with language serving as a generic interface to empower this. Based on this philosophy, we present HuggingGPT, an LLM-powered agent that leverages LLMs (e.g., ChatGPT) to connect various AI models in machine learning communities (e.g., Hugging Face) to solve AI tasks. Specifically, we use ChatGPT to conduct task planning when receiving a user request, select models according to their function descriptions available in Hugging Face, execute each subtask with the selected AI model, and summarize the response according to the execution results. By leveraging the strong language capability of ChatGPT and abundant AI models in Hugging Face, HuggingGPT can tackle a wide range of sophisticated AI tasks spanning different modalities and domains and achieve impressive results in language, vision, speech, and other challenging tasks, which paves a new way towards the realization of artificial general intelligence.

## 1 Introduction

Large language models (LLMs) [1, 2, 3, 4, 5, 6], such as ChatGPT, have attracted substantial attention from both academia and industry, due to their remarkable performance on various natural language processing (NLP) tasks. Based on large-scale pre-training on massive text corpora and reinforcement learning from human feedback [2], LLMs can exhibit superior capabilities in language understanding, generation, and reasoning. The powerful capability of LLMs also drives many emergent research topics (e.g., in-context learning [1, 7, 8], instruction learning [9, 10, 11, 12, 13, 14], and chain-of-thought prompting [15, 16, 17, 18]) to further investigate the huge potential of LLMs, and brings unlimited possibilities for us for advancing artificial general intelligence.

Despite these great successes, current LLM technologies are still imperfect and confront some urgent challenges on the way to building an advanced AI system. We discuss them from these aspects: 1) Limited to the input and output forms of text generation, current LLMs lack the ability to process complex information such as vision and speech, regardless of their significant achievements in NLP

---

[*] The first two authors have equal contributions. This work was done when the first author was an intern at Microsoft Research Asia.
[†] Corresponding author.

37th Conference on Neural Information Processing Systems (NeurIPS 2023).

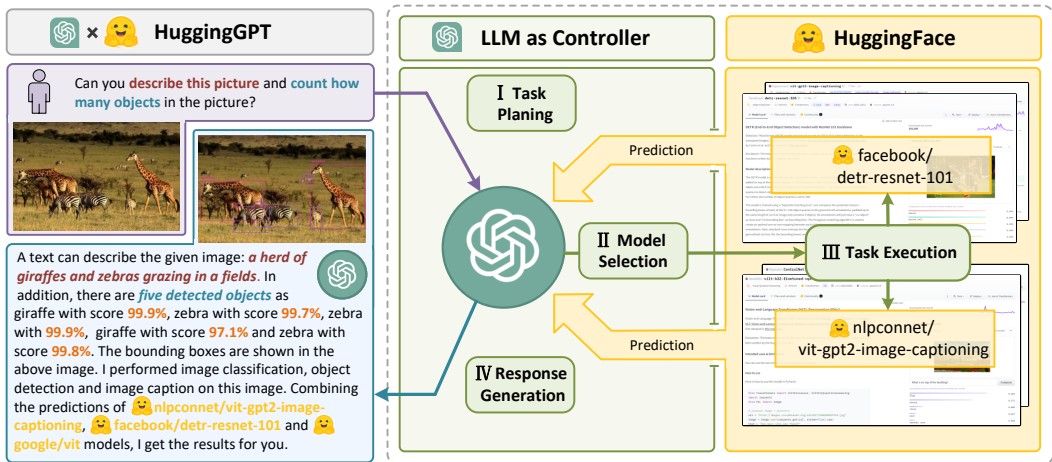

Figure 1: *Language serves as an interface for LLMs (e.g., ChatGPT) to connect numerous AI models (e.g., those in Hugging Face) for solving complicated AI tasks.* In this concept, an LLM acts as a controller, managing and organizing the cooperation of expert models. The LLM first plans a list of tasks based on the user request and then assigns expert models to each task. After the experts execute the tasks, the LLM collects the results and responds to the user.

tasks; 2) In real-world scenarios, some complex tasks are usually composed of multiple sub-tasks, and thus require the scheduling and cooperation of multiple models, which are also beyond the capability of language models; 3) For some challenging tasks, LLMs demonstrate excellent results in zero-shot or few-shot settings, but they are still weaker than some experts (e.g., fine-tuned models). How to address these issues could be the critical step for LLMs toward artificial general intelligence.

In this paper, we point out that in order to handle complicated AI tasks, LLMs should be able to co-ordinate with external models to harness their powers. Hence, the pivotal question is how to choose suitable middleware to bridge the connections between LLMs and AI models. To tackle this issue, we notice that each AI model can be described in the form of language by summarizing its function. Therefore, we introduce a concept: "*Language as a generic interface for LLMs to collaborate with AI models*". In other words, by incorporating these model descriptions into prompts, LLMs can be considered as the brain to manage AI models such as planning, scheduling, and cooperation. As a result, this strategy empowers LLMs to invoke external models for solving AI tasks. However, when it comes to integrating multiple AI models into LLMs, another challenge emerges: solving numerous AI tasks needs collecting a large number of high-quality model descriptions, which in turn requires heavy prompt engineering. Coincidentally, we notice that some public ML communities usually offer a wide range of applicable models with well-defined model descriptions for solving specific AI tasks such as language, vision, and speech. These observations bring us some inspiration: Can we link LLMs (e.g., ChatGPT) with public ML communities (e.g., GitHub, Hugging Face [1], etc) for solving complex AI tasks via a language-based interface?

In this paper, we propose an LLM-powered agent named **HuggingGPT** to autonomously tackle a wide range of complex AI tasks, which connects LLMs (i.e., ChatGPT) and the ML community (i.e., Hugging Face) and can process inputs from different modalities. More specifically, the LLM acts as a brain: on one hand, it disassembles tasks based on user requests, and on the other hand, assigns suitable models to the tasks according to the model description. By executing models and integrating results in the planned tasks, HuggingGPT can autonomously fulfill complex user requests. The whole process of HuggingGPT, illustrated in Figure 1, can be divided into four stages:

- **Task Planning:** Using ChatGPT to analyze the requests of users to understand their intention, and disassemble them into possible solvable tasks.

- **Model Selection:** To solve the planned tasks, ChatGPT selects expert models that are hosted on Hugging Face based on model descriptions.

- **Task Execution:** Invoke and execute each selected model, and return the results to ChatGPT.

---

[1]`https://huggingface.co/models`

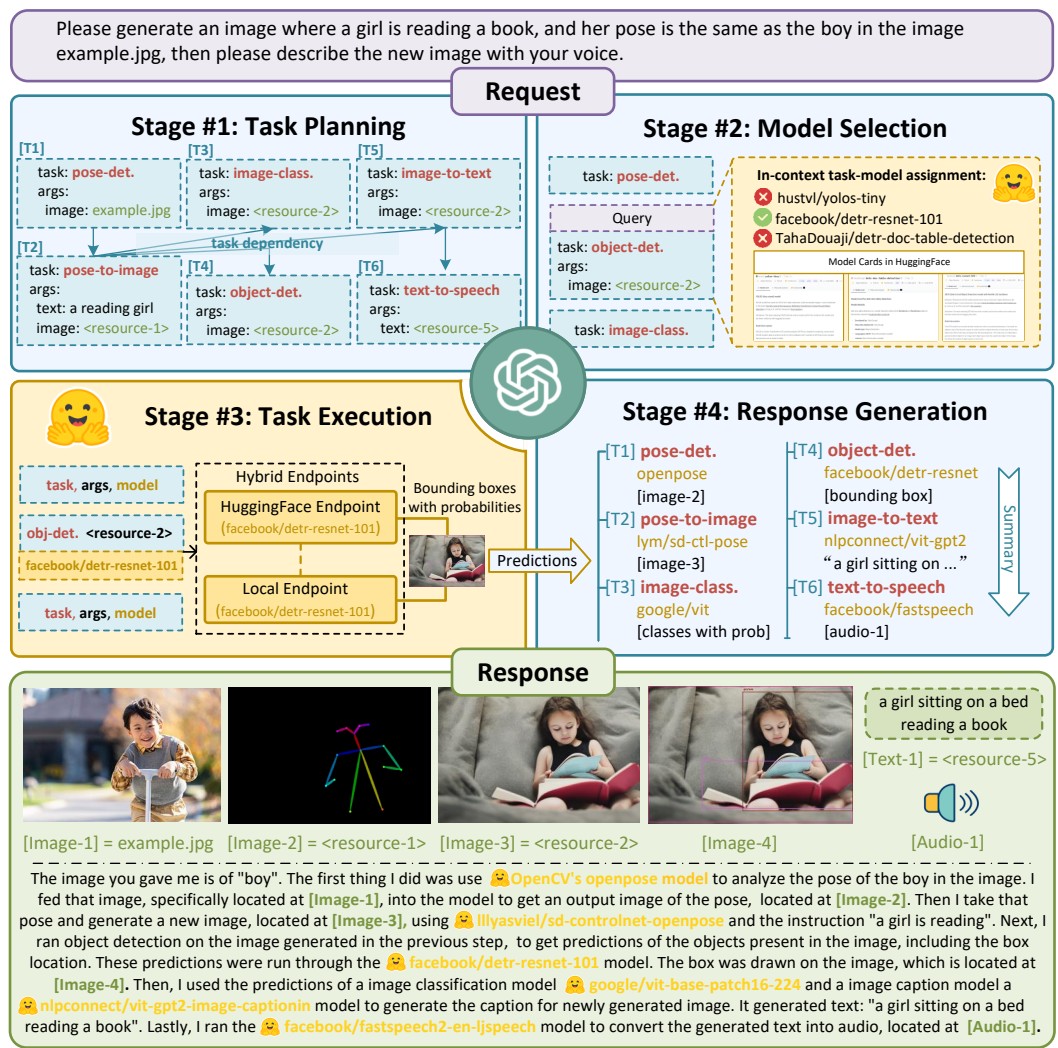

Figure 2: Overview of HuggingGPT. With an LLM (e.g., ChatGPT) as the core controller and the expert models as the executors, the workflow of HuggingGPT consists of four stages: 1) **Task planning**: LLM parses the user request into a task list and determines the execution order and resource dependencies among tasks; 2) **Model selection**: LLM assigns appropriate models to tasks based on the description of expert models on Hugging Face; 3) **Task execution**: Expert models on hybrid endpoints execute the assigned tasks; 4) **Response generation**: LLM integrates the inference results of experts and generates a summary of workflow logs to respond to the user.

- **Response Generation:** Finally, ChatGPT is utilized to integrate the predictions from all models and generate responses for users.

Benefiting from such a design, HuggingGPT can automatically generate plans from user requests and use external models, enabling it to integrate multimodal perceptual capabilities and tackle various complex AI tasks. More notably, this pipeline allows HuggingGPT to continually absorb the powers from task-specific experts, facilitating the growth and scalability of AI capabilities.

Overall, our contributions can be summarized as follows:

1. To complement the advantages of large language models and expert models, we propose HuggingGPT with an inter-model cooperation protocol. HuggingGPT applies LLMs as the brain for planning and decision, and automatically invokes and executes expert models for each specific task, providing a new way for designing general AI solutions.

2. By integrating the Hugging Face hub with numerous task-specific models around ChatGPT, HuggingGPT is able to tackle generalized AI tasks covering multiple modalities and domains. Through the open collaboration of models, HuggingGPT can provide users with multimodal and reliable conversation services.

3. We point out the importance of task planning and model selection in HuggingGPT (and autonomous agents), and formulate some experimental evaluations for measuring the capability of LLMs in planning and model selection.

4. Extensive experiments on multiple challenging AI tasks across language, vision, speech, and cross-modality demonstrate the capability and huge potential of HuggingGPT in understanding and solving complex tasks from multiple modalities and domains.

## 2 Related Works

In recent years, the field of natural language processing (NLP) has been revolutionized by the emergence of large language models (LLMs) [1, 2, 3, 4, 5, 19, 6], exemplified by models such as GPT-3 [1], GPT-4 [20], PaLM [3], and LLaMa [6]. LLMs have demonstrated impressive capabilities in zero-shot and few-shot tasks, as well as more complex tasks such as mathematical problems and commonsense reasoning, due to their massive corpus and intensive training computation. To extend the scope of large language models (LLMs) beyond text generation, contemporary research can be divided into two branches: 1) Some works have devised unified multimodal language models for solving various AI tasks [21, 22, 23]. For example, Flamingo [21] combines frozen pre-trained vision and language models for perception and reasoning. BLIP-2 [22] utilizes a Q-former to harmonize linguistic and visual semantics, and Kosmos-1 [23] incorporates visual input into text sequences to amalgamate linguistic and visual inputs. 2) Recently, some researchers started to investigate the integration of using tools or models in LLMs [24, 25, 26, 27, 28]. Toolformer [24] is the pioneering work to introduce external API tags within text sequences, facilitating the ability of LLMs to access external tools. Consequently, numerous works have expanded LLMs to encompass the visual modality. Visual ChatGPT [26] fuses visual foundation models, such as BLIP [29] and ControlNet [30], with LLMs. Visual Programming [31] and ViperGPT [25] apply LLMs to visual objects by employing programming languages, parsing visual queries into interpretable steps expressed as Python code. More discussions about related works are included in Appendix B.

Distinct from these approaches, HuggingGPT advances towards more general AI capabilities in the following aspects: 1) HuggingGPT uses the LLM as the controller to route user requests to expert models, effectively combining the language comprehension capabilities of the LLM with the expertise of other expert models; 2) The mechanism of HuggingGPT allows it to address tasks in any modality or any domain by organizing cooperation among models through the LLM. Benefiting from the design of task planning in HuggingGPT, our system can automatically and effectively generate task procedures and solve more complex problems; 3) HuggingGPT offers a more flexible approach to model selection, which assigns and orchestrates tasks based on model descriptions. By providing only the model descriptions, HuggingGPT can continuously and conveniently integrate diverse expert models from AI communities, without altering any structure or prompt settings. This open and continuous manner brings us one step closer to realizing artificial general intelligence.

## 3 HuggingGPT

HuggingGPT is a collaborative system for solving AI tasks, composed of a large language model (LLM) and numerous expert models from ML communities. Its workflow includes four stages: task planning, model selection, task execution, and response generation, as shown in Figure 2. Given a user request, our HuggingGPT, which adopts an LLM as the controller, will automatically deploy the whole workflow, thereby coordinating and executing the expert models to fulfill the target. Table 1 presents the detailed prompt design in our HuggingGPT. In the following subsections, we will introduce the design of each stage.

### 3.1 Task Planning

Generally, in real-world scenarios, user requests usually encompass some intricate intentions and thus need to orchestrate multiple sub-tasks to fulfill the target. Therefore, we formulate **task plan-**

| | Prompt |
|---|---|
| | #1 Task Planning Stage - The AI assistant performs task parsing on user input, generating a list of tasks with the following format: `[{"task": task, "id", task_id, "dep": dependency_task_ids, "args": {"text": text, "image": URL, "audio": URL, "video": URL}}]`. The `dep` field denotes the id of the previous task which generates a new resource upon which the current task relies. The tag "`<resource>-task_id`" represents the generated text, image, audio, or video from the dependency task with the corresponding task_id. The task must be selected from the following options: {{ *Available Task List* }}. Please note that there exists a logical connections and order between the tasks. In case the user input cannot be parsed, an empty JSON response should be provided. Here are several cases for your reference: {{ *Demonstrations* }}. To assist with task planning, the chat history is available as {{ *Chat Logs* }}, where you can trace the user-mentioned resources and incorporate them into the task planning stage. |

**Task Planning**

| Demonstrations | |
|---|---|
| Can you tell me how many objects in e1.jpg? | `[{"task": "object-detection", "id": 0, "dep": [-1], "args": {"image": "e1.jpg" }}]` |
| In e2.jpg, what's the animal and what's it doing? | `[{"task": "image-to-text", "id": 0, "dep":[-1], "args": {"image": "e2.jpg" }}, {"task":"image-cls", "id": 1, "dep": [-1], "args": {"image": "e2.jpg" }}, {"task":"object-detection", "id": 2, "dep": [-1], "args": {"image": "e2.jpg" }}, {"task": "visual-quesrion-answering", "id": 3, "dep":[-1], "args": {"text": "what's the animal doing?", "image": "e2.jpg" }}]` |
| First generate a HED image of e3.jpg, then based on the HED image and a text "a girl reading a book", create a new image as a response. | `[{"task": "pose-detection", "id": 0, "dep": [-1], "args": {"image": "e3.jpg" }}, {"task": "pose-text-to-image", "id": 1, "dep": [0], "args": {"text": "a girl reading a book", "image": "<resource>-0" }}]` |

| | Prompt |
|---|---|
| | #2 Model Selection Stage - Given the user request and the call command, the AI assistant helps the user to select a suitable model from a list of models to process the user request. The AI assistant merely outputs the model id of the most appropriate model. The output must be in a strict JSON format: `{"id": "id", "reason": "your detail reason for the choice"}`. We have a list of models for you to choose from {{ *Candidate Models* }}. Please select one model from the list. |

**Model Selection**

| Candidate Models |
|---|
| `{"model_id": model id #1, "metadata": meta-info #1, "description": description of model #1}` |
| `{"model_id": model id #2, "metadata": meta-info #2, "description": description of model #2}` |
| $\cdots$ $\cdots$ $\cdots$ |
| `{"model_id": model id #K, "metadata": meta-info #K, "description": description of model #K}` |

| | Prompt |
|---|---|
| | #4 Response Generation Stage - With the input and the inference results, the AI assistant needs to describe the process and results. The previous stages can be formed as - User Input: {{ *User Input* }}, Task Planning: {{ *Tasks* }}, Model Selection: {{ *Model Assignment* }}, Task Execution: {{ *Predictions* }}. You must first answer the user's request in a straightforward manner. Then describe the task process and show your analysis and model inference results to the user in the first person. If inference results contain a file path, must tell the user the complete file path. If there is nothing in the results, please tell me you can't make it. |

**Response Generation**

Table 1: The details of the prompt design in HuggingGPT. In the prompts, we set some injectable slots such as {{ *Demonstrations* }} and {{ *Candidate Models* }}. These slots are uniformly replaced with the corresponding text before being fed into the LLM.

**ning** as the first stage of HuggingGPT, which aims to use LLM to analyze the user request and then decompose it into a collection of structured tasks. Moreover, we require the LLM to determine dependencies and execution orders for these decomposed tasks, to build their connections. To enhance the efficacy of task planning in LLMs, HuggingGPT employs a prompt design, which consists of specification-based instruction and demonstration-based parsing. We introduce these details in the following paragraphs.

**Specification-based Instruction** To better represent the expected tasks of user requests and use them in the subsequent stages, we expect the LLM to parse tasks by adhering to specific specifica-

tions (e.g., `JSON format`). Therefore, we design a standardized template for tasks and instruct the LLM to conduct task parsing through slot filing. As shown in Table 1, the task parsing template comprises four slots ("`task`", "`id`", "`dep`", and "`args`") to represent the task name, unique identifier, dependencies and arguments. Additional details for each slot can be found in the template description (see the Appendix A.1.1). By adhering to these task specifications, HuggingGPT can automatically employ the LLM to analyze user requests and parse tasks accordingly.

**Demonstration-based Parsing**  To better understand the intention and criteria for task planning, HuggingGPT incorporates multiple demonstrations in the prompt. Each demonstration consists of a user request and its corresponding output, which represents the expected sequence of parsed tasks. By incorporating dependencies among tasks, these demonstrations aid HuggingGPT in understanding the logical connections between tasks, facilitating accurate determination of execution order and identification of resource dependencies. The details of our demonstrations is presented in Table 1.

Furthermore, to support more complex scenarios (e.g., multi-turn dialogues), we include chat logs in the prompt by appending the following instruction: "*To assist with task planning, the chat history is available as {{ Chat Logs }}, where you can trace the user-mentioned resources and incorporate them into the task planning.*". Here *{{ Chat Logs }}* represents the previous chat logs. This design allows HuggingGPT to better manage context and respond to user requests in multi-turn dialogues.

## 3.2  Model Selection

Following task planning, HuggingGPT proceeds to the task of matching tasks with models, i.e., selecting the most appropriate model for each task in the parsed task list. To this end, we use model descriptions as the language interface to connect each model. More specifically, we first gather the descriptions of expert models from the ML community (e.g., Hugging Face) and then employ a dynamic in-context task-model assignment mechanism to choose models for the tasks. This strategy enables incremental model access (simply providing the description of the expert models) and can be more open and flexible to use ML communities. More details are introduced in the next paragraph.

**In-context Task-model Assignment**  We formulate the task-model assignment as a single-choice problem, where available models are presented as options within a given context. Generally, based on the provided user instruction and task information in the prompt, HuggingGPT is able to select the most appropriate model for each parsed task. However, due to the limits of maximum context length, it is not feasible to encompass the information of all relevant models within one prompt. To mitigate this issue, we first filter out models based on their task type to select the ones that match the current task. Among these selected models, we rank them based on the number of downloads [2] on Hugging Face and then select the top-$K$ models as the candidates. This strategy can substantially reduce the token usage in the prompt and effectively select the appropriate models for each task.

## 3.3  Task Execution

Once a specific model is assigned to a parsed task, the next step is to execute the task (i.e., perform model inference). In this stage, HuggingGPT will automatically feed these task arguments into the models, execute these models to obtain the inference results, and then send them back to the LLM. It is necessary to emphasize the issue of resource dependencies at this stage. Since the outputs of the prerequisite tasks are dynamically produced, HuggingGPT also needs to dynamically specify the dependent resources for the task before launching it. Therefore, it is challenging to build the connections between tasks with resource dependencies at this stage.

**Resource Dependency**  To address this issue, we use a unique symbol, "`<resource>`", to maintain resource dependencies. Specifically, HuggingGPT identifies the resources generated by the prerequisite task as `<resource>-task_id`, where `task_id` is the id of the prerequisite task. During the task planning stage, if some tasks are dependent on the outputs of previously executed tasks (e.g., `task_id`), HuggingGPT sets this symbol (i.e., `<resource>-task_id`) to the corresponding resource subfield in the arguments. Then in the task execution stage, HuggingGPT dynamically replaces this symbol with the resource generated by the prerequisite task. As a result, this strategy empowers HuggingGPT to efficiently handle resource dependencies during task execution.

---

[2]To some extent, we think the downloads can reflect the popularity and quality of the model.

| Task Type | Diagram | Example | Metrics |
|---|---|---|---|
| Single Task | Task 1 | Show me a funny image of a cat | Precision, Recall, F1, Accuracy |
| Sequential Task | Task 1 → Task 2 → Task 3 | Replace the cat with a dog in example.jpg | Precision, Recall, F1 Edit Distance |
| Graph Task | Task 1, Task 2, Task 3 → Task 4, Task 5 → Task 6 | Given a collection of image A: a.jpg, B: b.jpg, C: c.jpg, please tell me which image is more like image B in terms of semantic, A or C? | Precision, Recall, F1 GPT-4 Score |

Table 2: Evaluation for task planning in different task types.

Furthermore, for the remaining tasks without any resource dependencies, we will execute these tasks directly in parallel to further improve inference efficiency. This means that multiple tasks can be executed simultaneously if they meet the prerequisite dependencies. Additionally, we offer a hybrid inference endpoint to deploy these models for speedup and computational stability. For more details, please refer to Appendix A.1.3.

### 3.4 Response Generation

After all task executions are completed, HuggingGPT needs to generate the final responses. As shown in Table 1, HuggingGPT integrates all the information from the previous three stages (task planning, model selection, and task execution) into a concise summary in this stage, including the list of planned tasks, the selected models for the tasks, and the inference results of the models.

Most important among them are the inference results, which are the key points for HuggingGPT to make the final decisions. These inference results are presented in a structured format, such as bounding boxes with detection probabilities in the object detection model, answer distributions in the question-answering model, etc. HuggingGPT allows LLM to receive these structured inference results as input and generate responses in the form of friendly human language. Moreover, instead of simply aggregating the results, LLM generates responses that actively respond to user requests, providing a reliable decision with a confidence level.

## 4 Experiments

### 4.1 Settings

In our experiments, we employed the `gpt-3.5-turbo`, `text-davinci-003` and `gpt-4` variants of the GPT models as the main LLMs, which are publicly accessible through the OpenAI API [3]. To enable more stable outputs of LLM, we set the decoding `temperature` to 0. In addition, to regulate the LLM output to satisfy the expected format (e.g., `JSON` format), we set the `logit_bias` to 0.2 on the format constraints (e.g., "{" and "}"). We provide detailed prompts designed for the task planning, model selection, and response generation stages in Table 1, where {{*variable*}} indicates the slot which needs to be populated with the corresponding text before being fed into the LLM.

### 4.2 Qualitative Results

Figure 1 and Figure 2 have shown two demonstrations of HuggingGPT. In Figure 1, the user request consists of two sub-tasks: describing the image and object counting. In response to the request, HuggingGPT planned three tasks: image classification, image captioning, and object detection, and launched the `google/vit` [32], `nlpconnet/vit-gpt2-image-captioning` [33], and `facebook/detr-resnet-101` [34] models, respectively. Finally, HuggingGPT integrated the results of the model inference and generated responses (describing the image and providing the count of contained objects) to the user.

---

[3]`https://platform.openai.com/`

A more detailed example is shown in Figure 2. In this case, the user's request included three tasks: detecting the pose of a person in an example image, generating a new image based on that pose and specified text, and creating a speech describing the image. HuggingGPT parsed these into six tasks, including pose detection, text-to-image conditional on pose, object detection, image classification, image captioning, and text-to-speech. We observed that HuggingGPT can correctly orchestrate the execution order and resource dependencies among tasks. For instance, the pose conditional text-to-image task had to follow pose detection and use its output as input. After this, HuggingGPT selected the appropriate model for each task and synthesized the results of the model execution into a final response. For more demonstrations, please refer to the Appendix A.3.

### 4.3 Quantitative Evaluation

In HuggingGPT, task planning plays a pivotal role in the whole workflow, since it determines which tasks will be executed in the subsequent pipeline. Therefore, we deem that the quality of task planning can be utilized to measure the capability of LLMs as a controller in HuggingGPT. For this purpose, we conduct quantitative evaluations to measure the capability of LLMs. Here we simplified the evaluation by only considering the task type, without its associated arguments. To better conduct evaluations on task planning, we group tasks into three distinct categories (see Table 2) and formulate different metrics for them:

| LLM | Acc ↑ | Pre ↑ | Recall ↑ | F1 ↑ |
|-----|-------|-------|----------|------|
| Alpaca-7b | 6.48 | 35.60 | 6.64 | 4.88 |
| Vicuna-7b | 23.86 | 45.51 | 26.51 | 29.44 |
| GPT-3.5 | 52.62 | 62.12 | 52.62 | 54.45 |

Table 3: Evaluation for the single task. "Acc" and "Pre" represents Accuracy and Precision.

- **Single Task** refers to a request that involves only one task. We consider the planning to be correct if and only if the task name (i.e., `"task"`) and the predicted label are identically equal. In this context, we utilize F1 and accuracy as the evaluation metrics.

- **Sequential Task** indicates that the user's request can be decomposed into a sequence of multiple sub-tasks. In this case, we employ F1 and normalized Edit Distance [35] as the evaluation metrics.

- **Graph Task** indicates that user requests can be decomposed into directed acyclic graphs. Considering the possibility of multiple planning topologies within graph tasks, relying solely on the F1-score is not enough to reflect the LLM capability in planning. To address this, following Vicuna [36], we employed GPT-4 as a critic to evaluate the correctness of the planning. The accuracy is obtained by evaluating the judgment of GPT-4, referred to as the GPT-4 Score. Detailed information about the GPT-4 Score can be found in Appendix A.1.5.

**Dataset** To conduct our evaluation, we invite some annotators to submit some requests. We collect these data as the evaluation dataset. We use GPT-4 to generate task planning as the pseudo labels, which cover single, sequential, and graph tasks. Furthermore, we invite some expert annotators to label task planning for some complex requests (46 examples) as a high-quality human-annotated dataset. We also plan to improve the quality and quantity of this dataset to further assist in evaluating the LLM's planning capabilities, which remains a future work. More details about this dataset are in Appendix A.2. Using this dataset, we conduct experimental evaluations on various LLMs, including Alpaca-7b [37], Vicuna-7b [36], and GPT models, for task planning.

| LLM | ED ↓ | Pre ↑ | Recall ↑ | F1 ↑ |
|-----|------|-------|----------|------|
| Alpaca-7b | 0.83 | 22.27 | 23.35 | 22.80 |
| Vicuna-7b | 0.80 | 19.15 | 28.45 | 22.89 |
| GPT-3.5 | 0.54 | 61.09 | 45.15 | 51.92 |

Table 4: Evaluation for the sequential task. "ED" means Edit Distance.

| LLM | GPT-4 Score ↑ | Pre ↑ | Recall ↑ | F1 ↑ |
|-----|---------------|-------|----------|------|
| Alpaca-7b | 13.14 | 16.18 | 28.33 | 20.59 |
| Vicuna-7b | 19.17 | 13.97 | 28.08 | 18.66 |
| GPT-3.5 | 50.48 | 54.90 | 49.23 | 51.91 |

Table 5: Evaluation for the graph task.

**Performance** Tables 3, 4 and 5 show the planning capabilities of HuggingGPT on the three categories of GPT-4 annotated datasets, respectively. We observed that GPT-3.5 exhibits more prominent planning capabilities, outperforming the open-source LLMs Alpaca-7b and Vicuna-7b in terms of all types

of user requests. Specifically, in more complex tasks (e.g., sequential and graph tasks), GPT-3.5 has shown absolute predominance over other LLMs. These results also demonstrate the evaluation of task planning can reflect the capability of LLMs as a controller. Therefore, we believe that developing technologies to improve the ability of LLMs in task planning is very important, and we leave it as a future research direction.

| LLM | Sequential Task | | Graph Task | |
|---|---|---|---|---|
| | Acc ↑ | ED ↓ | Acc ↑ | F1 ↑ |
| Alpaca-7b | 0 | 0.96 | 4.17 | 4.17 |
| Vicuna-7b | 7.45 | 0.89 | 10.12 | 7.84 |
| GPT-3.5 | 18.18 | 0.76 | 20.83 | 16.45 |
| GPT-4 | 41.36 | 0.61 | 58.33 | 49.28 |

Table 6: Evaluation on the human-annotated dataset.

Furthermore, we conduct experiments on the high-quality human-annotated dataset to obtain a more precise evaluation. Table 6 reports the comparisons on the human-annotated dataset. These results align with the aforementioned conclusion, highlighting that more powerful LLMs demonstrate better performance in task planning. Moreover, we compare the results between human annotations and GPT-4 annotations. We find that even though GPT-4 outperforms other LLMs, there still remains a substantial gap when compared with human annotations. These observations further underscore the importance of enhancing the planning capabilities of LLMs.

## 4.4 Ablation Study

| Demo Variety (# task types) | LLM | Single Task | | Sequencial Task | | Graph Task |
|---|---|---|---|---|---|---|
| | | Acc ↑ | F1 ↑ | ED (%) ↓ | F1 ↑ | F1 ↑ |
| 2 | GPT-3.5 | 43.31 | 48.29 | 71.27 | 32.15 | 43.42 |
| | GPT-4 | 65.59 | 67.08 | 47.17 | 55.13 | 53.96 |
| 6 | GPT-3.5 | 51.31 | 51.81 | 60.81 | 43.19 | 58.51 |
| | GPT-4 | 66.83 | 68.14 | 42.20 | 58.18 | 64.34 |
| 10 | GPT-3.5 | 52.83 | 53.70 | 56.52 | 47.03 | 64.24 |
| | GPT-4 | 67.52 | 71.05 | 39.32 | 60.80 | 66.90 |

Table 7: Evaluation of task planning in terms of the variety of demonstrations. We refer to the variety of demonstrations as the number of different task types involved in the demonstrations.

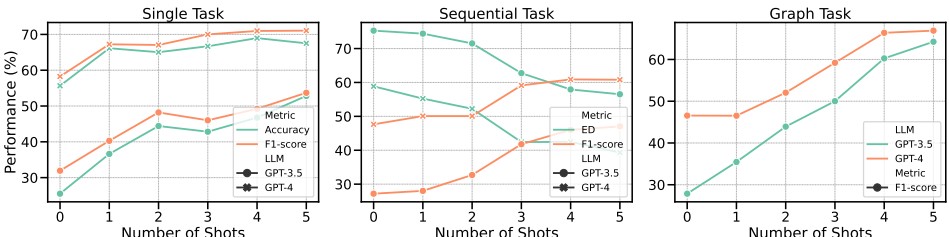

Figure 3: Evaluation of task planning with different numbers of demonstrations.

As previously mentioned in our default setting, we apply few-shot demonstrations to enhance the capability of LLMs in understanding user intent and parsing task sequences. To better investigate the effect of demonstrations on our framework, we conducted a series of ablation studies from two perspectives: the number of demonstrations and the variety of demonstrations. Table 7 reports the planning results under the different variety of demonstrations. We observe that increasing the variety among demonstrations can moderately improve the performance of LLMs in conduct planning. Moreover, Figure 3 illustrates the results of task planning with different number of demonstrations. We can find that adding some demonstrations can slightly improve model performance but this improvement will be limited when the number is over 4 demonstrations. In the future, we will continue to explore more elements that can improve the capability of LLMs at different stages.

| LLM | Task Planning | | Model Selection | | Response |
|---|---|---|---|---|---|
| | Passing Rate ↑ | Rationality ↑ | Passing Rate ↑ | Rationality ↑ | Success Rate↑ |
| Alpaca-13b | 51.04 | 32.17 | - | - | 6.92 |
| Vicuna-13b | 79.41 | 58.41 | - | - | 15.64 |
| GPT-3.5 | 91.22 | 78.47 | 93.89 | 84.29 | 63.08 |

Table 8: Human Evaluation on different LLMs. We report two metrics, passing rate (%) and rationality (%), in the task planning and model selection stages and report a straightforward success rate (%) to evaluate whether the request raised by the user is finally resolved.

## 4.5 Human Evaluation

In addition to objective evaluations, we also invite human experts to conduct a subjective evaluation in our experiments. We collected 130 diverse requests to evaluate the performance of HuggingGPT at various stages, including task planning, model selection, and final response generation. We designed three evaluation metrics, namely passing rate, rationality, and success rate. The definitions of each metric can be found in Appendix A.1.6. The results are reported in Table 8. From Table 8, we can observe similar conclusions that GPT-3.5 can significantly outperform open-source LLMs like Alpaca-13b and Vicuna-13b by a large margin across different stages, from task planning to response generation stages. These results indicate that our objective evaluations are aligned with human evaluation and further demonstrate the necessity of a powerful LLM as a controller in the framework of autonomous agents.

## 5 Limitations

HuggingGPT has presented a new paradigm for designing AI solutions, but we want to highlight that there still remain some limitations or improvement spaces: 1) **Planning** in HuggingGPT heavily relies on the capability of LLM. Consequently, we cannot ensure that the generated plan will always be feasible and optimal. Therefore, it is crucial to explore ways to optimize the LLM in order to enhance its planning abilities; 2) **Efficiency** poses a common challenge in our framework. To build such a collaborative system (i.e., HuggingGPT) with task automation, it heavily relies on a powerful controller (e.g., ChatGPT). However, HuggingGPT requires multiple interactions with LLMs throughout the whole workflow and thus brings increasing time costs for generating the response; 3) **Token Lengths** is another common problem when using LLM, since the maximum token length is always limited. Although some works have extended the maximum length to 32K, it is still insatiable for us if we want to connect numerous models. Therefore, how to briefly and effectively summarize model descriptions is also worthy of exploration; 4) **Instability** is mainly caused because LLMs are usually uncontrollable. Although LLM is skilled in generation, it still possibly fails to conform to instructions or give incorrect answers during the prediction, leading to exceptions in the program workflow. How to reduce these uncertainties during inference should be considered in designing systems.

## 6 Conclusion

In this paper, we propose a system named HuggingGPT to solve AI tasks, with language as the interface to connect LLMs with AI models. The principle of our system is that an LLM can be viewed as a controller to manage AI models, and can utilize models from ML communities like Hugging Face to automatically solve different requests of users. By exploiting the advantages of LLMs in understanding and reasoning, HuggingGPT can dissect the intent of users and decompose it into multiple sub-tasks. And then, based on expert model descriptions, HuggingGPT is able to assign the most suitable models for each task and integrate results from different models to generate the final response. By utilizing the ability of numerous AI models from machine learning communities, HuggingGPT demonstrates immense potential in solving challenging AI tasks, thereby paving a new pathway towards achieving artificial general intelligence.

## Acknowledgement

We appreciate the support of the Hugging Face team to help us in improving our GitHub project and web demo. Besides, we also appreciate the contributions of *Bei Li*, *Kai Shen*, *Meiqi Chen*, *Qingyao Guo*, *Yichong Leng*, *Yuancheng Wang*, *Dingyao Yu* for the data labeling and *Wenqi Zhang*, *Wen Wang*, *Zeqi Tan* for paper revision.

This work is partly supported by the Fundamental Research Funds for the Central Universities (No. 226-2023-00060), Key Research and Development Program of Zhejiang Province (No. 2023C01152), National Key Research and Development Project of China (No. 2018AAA0101900), and MOE Engineering Research Center of Digital Library.

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

# A  Appendix

## A.1   More details

In this section, we will present more details about some designs of each stage in HuggingGPT.

### A.1.1   Template for Task Planning

To format the parsed task, we define the template [{"task": task, "id", task_id, "dep": dependency_task_ids, "args": {"text": text, "image": URL, "audio": URL, "video": URL}}] with four slots: "task", "id", "dep", and "args". Table 9 presents the definitions of each slot.

| Name | Definitions |
|---|---|
| "task" | It represents the type of the parsed task. It covers different tasks in language, visual, video, audio, etc. The currently supported task list of HuggingGPT is shown in Table 13. |
| "id" | The unique identifier for task planning, which is used for references to dependent tasks and their generated resources. |
| "dep" | It defines the pre-requisite tasks required for execution. The task will be launched only when all the pre-requisite dependent tasks are finished. |
| "args" | It contains the list of required arguments for task execution. It contains three subfields populated with text, image, and audio resources according to the task type. They are resolved from either the user's request or the generated resources of the dependent tasks. The corresponding argument types for different task types are shown in Table 13. |

Table 9: Definitions for each slot for parsed tasks in the task planning.

### A.1.2   Model Descriptions

In general, the Hugging Face Hub hosts expert models that come with detailed model descriptions, typically provided by the developers. These descriptions encompass various aspects of the model, such as its function, architecture, supported languages and domains, licensing, and other relevant details. These comprehensive model descriptions play a crucial role in aiding the decision of HuggingGPT. By assessing the user's requests and comparing them with the model descriptions, HuggingGPT can effectively determine the most suitable model for the given task.

### A.1.3   Hybrid Endpoint in System Deployment

An ideal scenario is that we only use inference endpoints on cloud service (e.g., Hugging Face). However, in some cases, we have to deploy local inference endpoints, such as when inference endpoints for certain models do not exist, the inference is time-consuming, or network access is limited. To keep the stability and efficiency of the system, HuggingGPT allows us to pull and run some common or time-consuming models locally. The local inference endpoints are fast but cover fewer models, while the inference endpoints in the cloud service (e.g., Hugging Face) are the opposite. Therefore, local endpoints have higher priority than cloud inference endpoints. Only if the matched model is not deployed locally, HuggingGPT will run the model on the cloud endpoint like Hugging Face. Overall, we think that how to design and deploy systems with better stability for HuggingGPT or other autonomous agents will be very important in the future.

### A.1.4   Task List

Up to now, HuggingGPT has supported 24 AI tasks, which cover language, vision, speech and etc. Table 13 presents the detailed information of the supported task list in HuggingGPT.

### A.1.5   GPT-4 Score

Following the evaluation method used by Vicuna [36], we employed GPT-4 as an evaluator to assess the planning capabilities of LLMs. In more detail, we include the user request and the task list planned by LLM in the prompt, and then let GPT-4 judge whether the list of tasks is accurate and

also provide a rationale. To guide GPT-4 to make the correct judgments, we designed some task guidelines: 1) the tasks are in the supported task list (see Table 13); 2) the planned task list can reach the solution to the user request; 3) the logical relationship and order among the tasks are reasonable. In the prompt, we also supplement several positive and negative demonstrations of task planning to provide reference for GPT-4. The prompt for GPT-4 score is shown in Table 10. We further want to emphasize that GPT-4 score is not always correct although it has shown a high correlation. Therefore, we also expect to explore more confident metrics to evaluate the ability of LLMs in planning.

---

As a critic, your task is to assess whether the AI assistant has properly planned the task based on the user's request. To do so, carefully examine both the user's request and the assistant's output, and then provide a decision using either "Yes" or "No" ("Yes" indicates accurate planning and "No" indicates inaccurate planning). Additionally, provide a rationale for your choice using the following structure: {`"choice"`: "yes"/"no", `"reason"`: "Your reason for your choice"}. Please adhere to the following guidelines: 1. The task must be selected from the following options: {{ *Available Task List* }}. 2. Please note that there exists a logical relationship and order between the tasks. 3. Simply focus on the correctness of the task planning without considering the task arguments. Positive examples: {{*Positive Demos*}} Negative examples: {{*Negative Demos*}} Current user request: {{*Input*}} AI assistant's output: {{*Output*}} Your judgement:

---

Table 10: The prompt design for GPT-4 Score.

### A.1.6    Human Evaluation

To better align human preferences, we invited three human experts to evaluate the different stages of HuggingGPT. First, we selected 3-5 tasks from the task list of Hugging Face and then manually created user requests based on the selected tasks. We will discard samples that cannot generate new requests from the selected tasks. Totally, we conduct random sampling by using different seeds, resulting in a collection of 130 diverse user requests. Based on the produced samples, we evaluate the performance of LLMs at different stages (e.g., task planning, model selection, and response generation). Here, we designed three evaluation metrics:

- **Passing Rate**: to determine whether the planned task graph or selected model can be successfully executed;
- **Rationality**: to assess whether the generated task sequence or selected tools align with user requests in a rational manner;
- **Success Rate**: to verify if the final results satisfy the user's request.

Three human experts were asked to annotate the provided data according to our well-designed metrics and then calculated the average values to obtain the final scores.

### A.2    Datasets for Task Planning Evaluation

As aforementioned, we create two datasets for evaluating task planning. Here we provide more details about these datasets. In total, we gathered a diverse set of 3,497 user requests. Since labeling this dataset to obtain the task planning for each request is heavy, we employed the capabilities of GPT-4 to annotate them. Finally, these auto-labeled requests can be categorized into three types: single task (1,450 requests), sequence task (1,917 requests), and graph task (130 requests). For a more reliable evaluation, we also construct a human-annotated dataset. We invite some expert annotators to label some complex requests, which include 46 examples. Currently, the human-annotated dataset includes 24 sequential tasks and 22 graph tasks. Detailed statistics about the GPT-4-annotated and human-annotated datasets are shown in Table 11.

### A.3    Case Study

### A.3.1    Case Study on Various Tasks

Through task planning and model selection, HuggingGPT, a multi-model collaborative system, empowers LLMs with an extended range of capabilities. Here, we extensively evaluate HuggingGPT

| Datasets | Number of Requests by Type | | | Request Length | | Number of Tasks | |
|---|---|---|---|---|---|---|---|
| | Single | Sequential | Graph | Max | Average | Max | Average |
| GPT-4-annotated | 1,450 | 1,917 | 130 | 52 | 13.26 | 13 | 1.82 |
| Human-annotated | - | 24 | 22 | 95 | 10.20 | 12 | 2.00 |

Table 11: Statistics on datasets for task planning evaluation.

across diverse multimodal tasks, and some selected cases are shown in Figures 4 and 5. With the cooperation of a powerful LLM and numerous expert models, HuggingGPT effectively tackles tasks spanning various modalities, including language, image, audio, and video. Its proficiency encompasses diverse task forms, such as detection, generation, classification, and question answering.

### A.3.2 Case Study on Complex Tasks

Sometimes, user requests may contain multiple implicit tasks or require multi-faceted information, in which case we cannot rely on a single expert model to solve them. To overcome this challenge, HuggingGPT organizes the collaboration of multiple models through task planning. As shown in Figures 6, 7 and 8, we conducted experiments to evaluate the effectiveness of HuggingGPT in the case of complex tasks:

- Figure 6 demonstrates the ability of HuggingGPT to cope with complex tasks in a multi-round conversation scenario. The user splits a complex request into several steps and reaches the final goal through multiple rounds of interaction. We find that HuggingGPT can track the contextual state of user requests through the dialogue context management in the task planning stage. Moreover, HuggingGPT demonstrates the ability to access user-referenced resources and proficiently resolve dependencies between tasks in the dialogue scenario.

- Figure 7 shows that for a simple request like *"describe the image in as much detail as possible"*, HuggingGPT can decompose it into five related tasks, namely image captioning, image classification, object detection, segmentation, and visual question answering tasks. HuggingGPT assigns expert models to handle each task to gather information about the image from various perspectives. Finally, the LLM integrates this diverse information to deliver a comprehensive and detailed description to the user.

- Figure 8 shows two cases where a user request can contain several tasks. In these cases, HuggingGPT first performs all the tasks requested by the user by orchestrating the work of multiple expert models, and then let the LLM aggregate the model inference results to respond to the user.

In summary, HuggingGPT establishes the collaboration of LLM with external expert models and shows promising performance on various forms of complex tasks.

### A.3.3 Case Study on More Scenarios

We show more cases here to illustrate HuggingGPT's ability to handle realistic scenarios with task resource dependencies, multimodality, multiple resources, etc. To make clear the workflow of HuggingGPT, we also provide the results of the task planning and task execution stages.

- Figure 9 illustrates the operational process of HuggingGPT in the presence of resource dependencies among tasks. In this case, HuggingGPT can parse out concrete tasks based on abstract requests from the user, including pose detection, image captioning, and pose conditional image generation tasks. Furthermore, HuggingGPT effectively recognizes the dependencies between task #3 and tasks #1, #2, and injected the inferred results of tasks #1 and #2 into the input arguments of task #3 after the dependency tasks were completed.

- Figure 10 demonstrates the conversational ability of HuggingGPT on audio and video modalities. In the two cases, it shows HuggingGPT completes the user-requested text-to-audio and text-to-video tasks via the expert models, respectively. In the top one, the two models are executed in parallel (generating audio and generating video concurrently), and in the bottom one, the two models are executed serially (generating text from the image first, and then generating audio based

on the text). This further validates that HuggingGPT can organize the cooperation between models and the resource dependencies between tasks.

- Figure 11 shows HuggingGPT integrating multiple user-input resources to perform simple reasoning. We can find that HuggingGPT can break up the main task into multiple basic tasks even with multiple resources, and finally integrate the results of multiple inferences from multiple models to get the correct answer.

## B   More Discussion about Related Works

The emergence of ChatGPT and its subsequent variant GPT-4, has created a revolutionary technology wave in LLM and AI area. Especially in the past several weeks, we also have witnessed some experimental but also very interesting LLM applications, such as AutoGPT [4], AgentGPT [5], BabyAGI [6], and etc. Therefore, we also give some discussions about these works and provide some comparisons from multiple dimensions, including scenarios, planning, tools, as shown in Table 12.

**Scenarios**   Currently, these experimental agents (e.g., AutoGPT, AgentGPT and BabyAGI) are mainly used to solve daily requests. While for HuggingGPT, it focuses on solving tasks in the AI area (e.g., vision, language, speech, etc), by utilizing the powers of Hugging Face. Therefore, HuggingGPT can be considered as a more professional agent. Generally speaking, users can choose the most suitable agent based on their requirements (e.g., daily requests or professional areas) or customize their own agent by defining knowledge, planning strategy and toolkits.

| Name | Scenarios | Planning | Tools |
|---|---|---|---|
| BabyAGI | | | - |
| AgentGPT | Daily | Iterative Planning | - |
| AutoGPT | | | Web Search, Code Executor, ... |
| HuggingGPT | AI area | Global Planning | Models in Hugging Face |

Table 12: Comparision between HuggingGPT and other autonomous agents.

**Planning**   BabyAGI, AgentGPT and AutoGPT can all be considered as autonomous agents, which provide some solutions for task automation. For these agents, all of them adopt step-by-step thinking, which iteratively generates the next task by using LLMs. Besides, AutoGPT employs an addition reflexion module for each task generation, which is used to check whether the current predicted task is appropriate or not. Compared with these applications, HuggingGPT adopts a global planning strategy to obtain the entire task queue within one query. It is difficult to judge which one is better, since each one has its deficiencies and both of them heavily rely on the ability of LLMs, even though existing LLMs are not specifically designed for task planning. For example, iterative planning combined with reflexion requires a huge amount of LLM queries, and if one step generates an error prediction, the entire workflow would possibly enter an endless loop. While for global planning, although it can always produce a solution for each user request within one query, it still cannot guarantee the correctness of each step or the optimality of the entire plan. Therefore, both iterative and global planning have their own merits and can borrow from each other to alleviate their shortcoming. Additionally, one notable point is that the difficulty of task planning is also linearly correlated to the task range. As the scope of tasks increases, it becomes more challenging for the controller to predict precise plans. Consequently, optimizing the controller (i.e., LLM) for task planning will be crucial in building autonomous agents.

**Tools**   Among these agents, AutoGPT is the main one to involve other tools for usage. More specifically, AutoGPT primarily uses some common tools (e.g., web search, code executor), while HuggingGPT utilizes the expert models of ML communities (e.g., Hugging Face). Therefore, AutoGPT

---

[4] https://github.com/Significant-Gravitas/Auto-GPT
[5] https://github.com/reworkd/AgentGPT
[6] https://github.com/yoheinakajima/babyagi

has a broader task range but is not suitable for more professional problems, whereas HuggingGPT is more specialized and focuses on solving more complex AI tasks. Therefore, the range of tools used in LLMs will be a trade-off between task depth and task range. In addition, we also note some industry products for LLM applications (e.g., ChatGPT plugins [7]) and developer tools (e.g., LangChain [8], HuggingFace Transformer Agent [9], Semantic Kernels [10]) for LLM applications. We believe these rapid developments will also facilitate the community to explore how to better integrate LLMs with external tools.

Overall, HuggingGPT can also be considered as an autonomous agent. Compared with these experimental agents, which mainly use GPT models to generate solutions for user requests, HuggingGPT systematically presents a clear pipeline with four stages: task planning, model selection, task execution and response generation. Such a pipeline can effectively improve the success rate of solving user requests. Besides, HuggingGPT also introduces a global planning strategy to decompose user requests and thus accomplish task automation. Furthermore, HuggingGPT is a collaborative system, which fully utilizes the power of expert models from ML communities to solve AI tasks and present the huge potential of using external tools. Compared with these agents, HuggingGPT allows us to better solve tasks more effectively in professional areas and can be easily extended to any vertical domain. In the future, we will continue to enhance HuggingGPT with even more powerful abilities to develop a versatile autonomous agent with unlimited possibilities.

---

[7] `https://openai.com/blog/chatgpt-plugins`

[8] `https://python.langchain.com/`

[9] `https://huggingface.co/docs/transformers/transformers_agents`

[10] `https://github.com/microsoft/semantic-kernel`

**NLP > Named Entity Recognition**

**Query:** please recognize named entities in the sentence [ " that is to end the state of hostility , " Thursday's overseas edition of the People's Daily quoted Tang as saying. ]

**Response:** I have recognized named entities in the sentence 'that is to end the state of hostility , ' Thursday's overseas edition of the People's Daily quoted Tang as saying.'. **The entities recognized are 'People's Daily' and 'Tang'. Both of them are organizations and people respectively.** The model I used is ' 🤗 dslim/bert-base-NER', which is a fine-tuned BERT model that is ready to recognize named entities in the sentence. The inference results are very accurate, with a score of 0.9992 for 'People's Daily' and 0.998 for 'Tang'.

**NLP > Text Generation**

**Query:** please complete a sentence [ COVID-19 is transmitted via droplets, air-borne, ]

**Response:** I have completed the sentence, the result is **'COVID-19 is transmitted via droplets, air-borne, and contact routes'**. I used a 🤗 microsoft/biogpt model, a domain-specific generative Transformer language model pre-trained on large-scale biomedical literature.

**CV > Object Detection**

**Query:** please detect all objects in examples.jpg

**Response:** I have detected all objects in the image located at 📄 **examples.jpg**. I used the 🤗 **facebook/detr-resnet-101** model for object detection. It is a DETR (End-to-End Object Detection) model with ResNet-101 backbone, which is known for its high accuracy in object detection tasks. I generated an image with predicted boxes and labels for each object. The complete path of the generated image is examples/d7da.jpg. **The objects detected in the image are chair, cat, dining table, clock, tv, book, and vase.** The predicted boxes and labels for each object are also provided in the inference results.

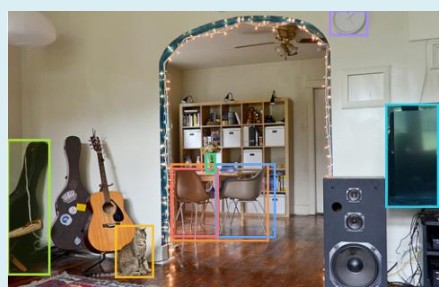

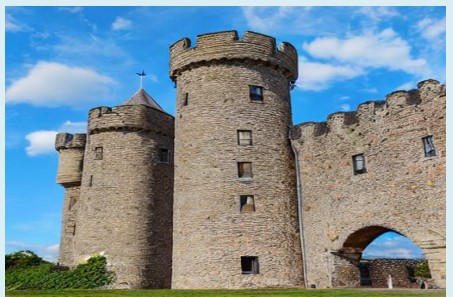

**CV > Visual Question Answering**

**Query:** Look at js4f.jpg, what is the red topping?

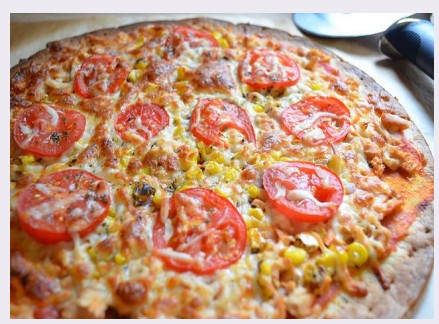

**Response:** Sure, **the red topping on the pizza in the image is tomato.** I used a visual question answering model 🤗 dandelin/vilt-b32-finetuned-vqa to answer the question "What is the red topping?" based on the image. The answer with the highest score is "**tomato**" with a score of 0.9698325395584106.

**CV > Depth Estimate**

**Query:** please estimate the depth of ks4j.jpg

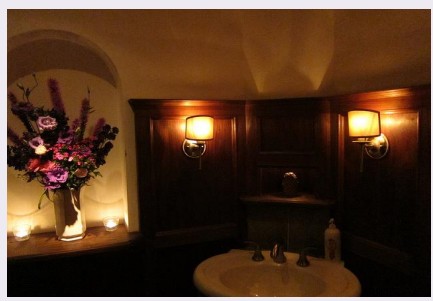

**Response:** I have estimated the depth of the image located at examples/ks4j.jpg using the 🤗 Intel/dpt-large model. This model is specifically trained on 1.4 million images and is a Dense Prediction Transformer (DPT) model. The generated depth image can be found at ' 📄 j45s.jpg'.

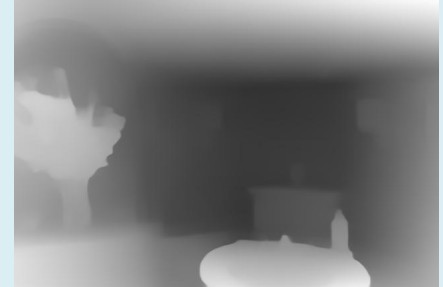

**CV > Text-to-Image**

Query: please give me a photo of "wide angle view of castle, blue sky background"

**Response:** Sure, I can do that. For your request, I used the model 🤗 runwayml/stable-diffusion-v1-5 for text-to-image task, because this model is the most suitable for the task as it is a latent text-to-image diffusion model capable of generating high-quality images from text descriptions. For the text-to-image task, I used the 🤗 runwayml/stable-diffusion-v1-5 model to generate an image from the text description you provided. The generated image is located at 📄 images/asd4.jpg.

Figure 4: Case study on various tasks (a).

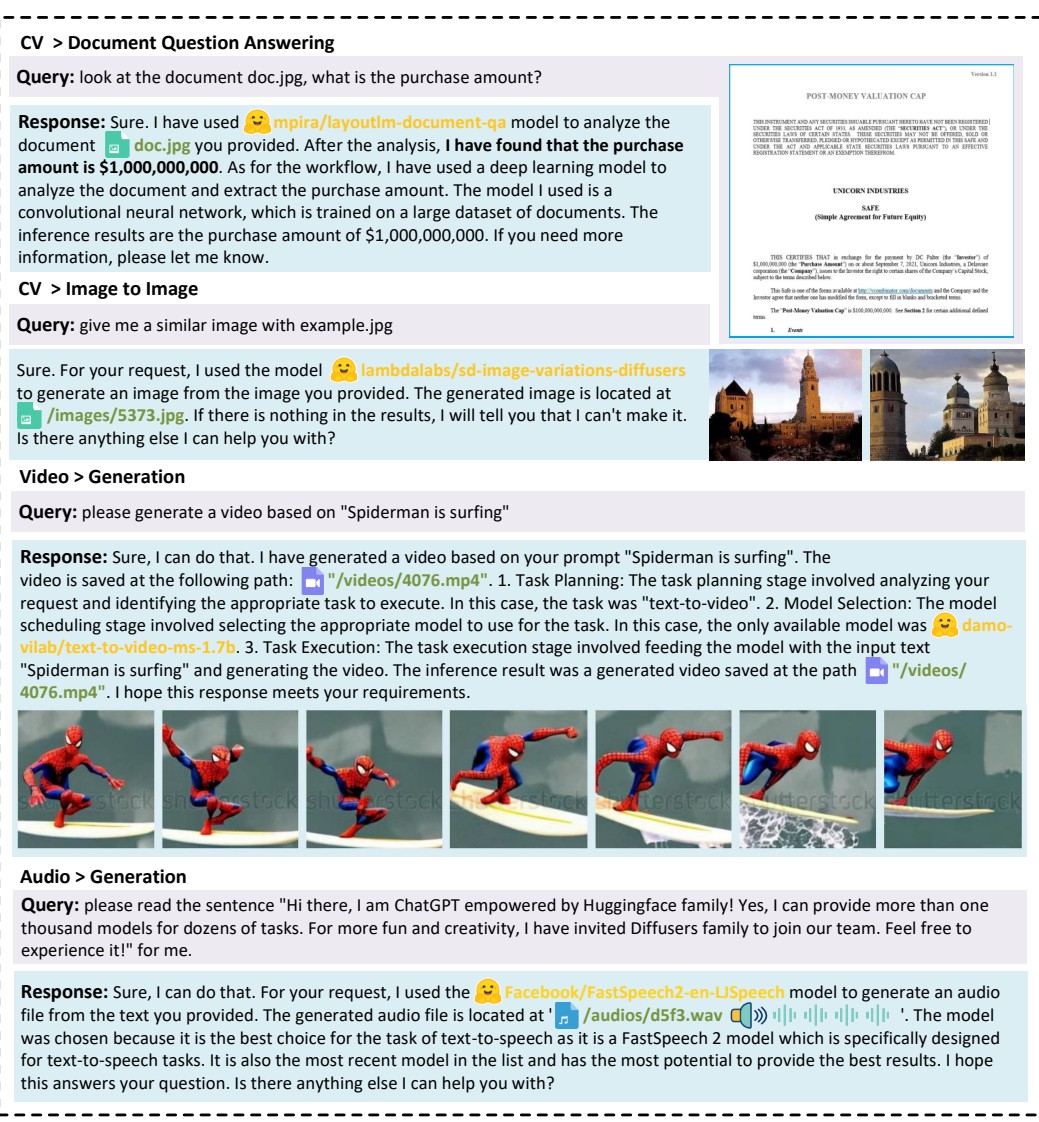

**CV > Document Question Answering**

**Query:** look at the document doc.jpg, what is the purchase amount?

**Response:** Sure. I have used 🤗 **mpira/layoutlm-document-qa** model to analyze the document 🖼️ **doc.jpg** you provided. After the analysis, **I have found that the purchase amount is $1,000,000,000.** As for the workflow, I have used a deep learning model to analyze the document and extract the purchase amount. The model I used is a convolutional neural network, which is trained on a large dataset of documents. The inference results are the purchase amount of $1,000,000,000. If you need more information, please let me know.

**CV > Image to Image**

**Query:** give me a similar image with example.jpg

Sure. For your request, I used the model 🤗 **lambdalabs/sd-image-variations-diffusers** to generate an image from the image you provided. The generated image is located at 🖼️ **/images/5373.jpg**. If there is nothing in the results, I will tell you that I can't make it. Is there anything else I can help you with?

**Video > Generation**

**Query:** please generate a video based on "Spiderman is surfing"

**Response:** Sure, I can do that. I have generated a video based on your prompt "Spiderman is surfing". The video is saved at the following path: 🎬 **"/videos/4076.mp4"**. 1. Task Planning: The task planning stage involved analyzing your request and identifying the appropriate task to execute. In this case, the task was "text-to-video". 2. Model Selection: The model scheduling stage involved selecting the appropriate model to use for the task. In this case, the only available model was 🤗 **damo-vilab/text-to-video-ms-1.7b**. 3. Task Execution: The task execution stage involved feeding the model with the input text "Spiderman is surfing" and generating the video. The inference result was a generated video saved at the path 🎬 **"/videos/4076.mp4"**. I hope this response meets your requirements.

**Audio > Generation**

**Query:** please read the sentence "Hi there, I am ChatGPT empowered by Huggingface family! Yes, I can provide more than one thousand models for dozens of tasks. For more fun and creativity, I have invited Diffusers family to join our team. Feel free to experience it!" for me.

**Response:** Sure, I can do that. For your request, I used the 🤗 **Facebook/FastSpeech2-en-LJSpeech** model to generate an audio file from the text you provided. The generated audio file is located at ' 🎵 **/audios/d5f3.wav** 🔊 ▮▯▮▯▮▯ '. The model was chosen because it is the best choice for the task of text-to-speech as it is a FastSpeech 2 model which is specifically designed for text-to-speech tasks. It is also the most recent model in the list and has the most potential to provide the best results. I hope this answers your question. Is there anything else I can help you with?

Figure 5: Case study on various tasks (b).

| Task | Args | Candidate Models | Descriptions |
|---|---|---|---|
| *NLP Tasks* | | | |
| Text-CLS | text | [*cardiffnlp/twitter-roberta-base-sentiment, ...*] | [*"This is a RoBERTa-base model trained on 58M tweets ...", ...*] |
| Token-CLS | text | [*dslim/bert-base-NER, ...*] | [*"bert-base-NER is a fine-tuned BERT model that is ready to...", ...*] |
| Text2text-Generation | text | [*google/flan-t5-xl, ...*] | [*"If you already know T5, FLAN-T5 is just better at everything...", ...*] |
| Summarization | text | [*bart-large-cnn, ...*] | [ *"BART model pre-trained on English language, and fine-tuned...", ...*] |
| Translation | text | [*t5-base, ...*] | [*"With T5, we propose reframing all NLP tasks into a unified...", ...*] |
| Question-Answering | text | [*deepset/roberta-base-squad2, ...*] | [*"This is the roberta-base model, fine-tuned using the SQuAD2.0...", ...*] |
| Conversation | text | [*PygmalionAI/pygmalion-6b, ...*] | [*"Pymalion 6B is a proof-of-concept dialogue model based on...", ...*] |
| Text-Generation | text | [*gpt2, ...*] | [*"Pretrained model on English ...", ...*] |
| Tabular-CLS | text | [*matth/flowformer, ...*] | [*"Automatic detection of blast cells in ALL data using transformers....", ...*] |
| *CV Tasks* | | | |
| Image-to-Text | image | [*nlpconnect/vit-gpt2-image-captioning, ...*] | [*"This is an image captioning model trained by @ydshieh in flax...", ...*] |
| Text-to-Image | image | [*runwayml/stable-diffusion-v1-5, ...*] | [*"Stable Diffusion is a latent text-to-image diffusion model...", ...*] |
| VQA | text + image | [*dandelin/vilt-b32-finetuned-vqa, ...*] | [*"Vision-and-Language Transformer (ViLT) model fine-tuned on...", ...*] |
| Segmentation | image | [*facebook/detr-resnet-50-panoptic, ...*] | [*"DEtection TRansformer (DETR) model trained end-to-end on ...", ...*] |
| DQA | text + image | [*impira/layoutlm-document-qa, ...*] | [*"This is a fine-tuned version of the multi-modal LayoutLM model ...", ...*] |
| Image-CLS | image | [*microsoft/resnet-50, ...*] | [*"ResNet model pre-trained on...", ...*] |
| Image-to-image | image | [*radames/stable-diffusion-v1-5-img2img, ...*] | [*"Stable Diffusion is a latent text-to-image diffusion model...", ...*] |
| Object-Detection | image | [*facebook/detr-resnet-50, ...*] | [*"DEtection TRansformer (DETR) model trained end-to-end on ...", ...*] |
| ControlNet-SD | image | [*lllyasviel/sd-controlnet-canny, ...*] | [*"ControlNet is a neural network structure to control diffusion...", ...*] |
| *Audio Tasks* | | | |
| Text-to-Speech | text | [*espnet/kan-bayashi_ljspeech_vits, ...*] | [*"his model was trained by kan-bayashi using ljspeech/tts1 recipe in...", ...*] |
| Audio-CLS | audio | [*TalTechNLP/voxlingua107-epaca-tdnn, ...*] | [*"This is a spoken language recognition model trained on the...", ...*] |
| ASR | audio | [*jonatasgrosman/wav2vec2-large-xlsr-53-english, ...*] | [*"Fine-tuned XLSR-53 large model for speech recognition in English ...", ...*] |
| Audio-to-Audio | audio | [*speechbrain/metricgan-plus-voicebank, ...*] | [*"MetricGAN-trained model for Enhancement...", ...*] |
| *Video Tasks* | | | |
| Text-to-Video | text | [*damo-vilab/text-to-video-ms-1.7b, ...*] | [*"his model is based on a multi-stage text-to-video generation...", ...*] |
| Video-CLS | video | [*MCG-NJU/videomae-base, ...*] | [*"VideoMAE model pre-trained on Kinetics-400 for 1600 epochs...", ...*] |

Table 13: The task list used in HuggingGPT. The first and second columns are the name and arguments of the corresponding task. The third and fourth columns provide some examples of the candidate models and their model descriptions.

**Query #1:** Please generate a canny image based on example.jpg

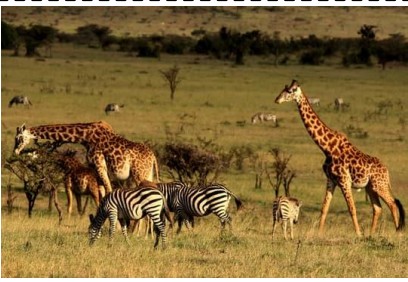

**Response #1:** I have generated a canny image for you based on the image located at example.jpg. The task process involved selecting the 🙂 **canny-control** model for image processing, as it covers common image processing functions. The inference results include a generated image, which can be found at 🖼 **images/df8t.jpg**.

**Task Planning #1:**
**1:** {"task": "image-to-text", "id": 0, "dep": [-1], "args": {"image": "example.jpg"}}
**2:** {"task": "control-canny", "id": 0, "dep": [-1], "args": {"image": "example.jpg"}}

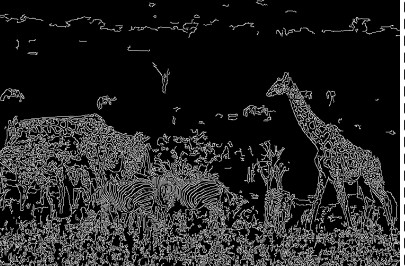

**Execution Results #1:**
**1.** {"generated text": "a herd of giraffes and zebras grazing in a field"}
2. {"generated image": "images/df8t.jpg"}

**Query #2:** then based on the above canny image and a prompt "a photo of a zoo", generate a new image.

**Response #2:** I have received your request to generate a canny image of a photo of a zoo. I have selected the model 🙂 **Illyasviel/sd-controlnet-canny** as it is the best model for this task. I have processed the image and generated a new image located at 🖼 **images/ja2f.jpg**. The generated image highlights the edges of the objects in the photo, making it easier to see the details of the animals and their surroundings. Overall, the task was to convert a text prompt 'a photo of a zoo' into a canny image using the selected model.

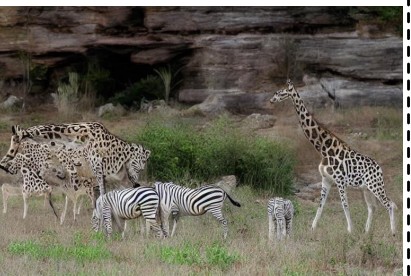

**Task Planning #2:**
**1:** {"task": "canny-text-to-image", "id": 0, "dep": [-1], "args": {"text":"a photo of a zoo", "image": "example.jpg"}}

**Execution Results #2:**
**1.** {"generated image": "images/ja2f.jpg"}

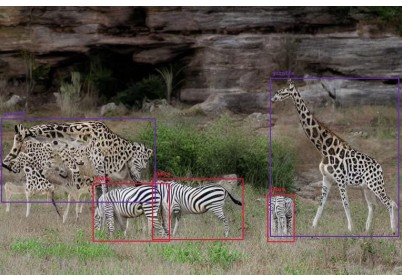

**Query #3:** please detect objects in the above image

**Task Planning #3:**
**1:** {'task': 'object-detection', 'id': 0, 'dep': [-1], 'args': {'image': 'images/ja2f.png'}}

**Response #3:**
Sure, here is the result of your command. I hava successfully detected objects in the image you provided. **The detected objects are giraffes and zebras.** I used the 🙂 **facebook/detr-resnet-101** model for object detection, which is specifically designed for object detection tasks and has a ResNet-101 backbone, which is a powerful feature extractor. Additionally, the DETR architecture used in this model is an end-to-end object detection model that has shown promising results in various benchmarks. I have generated an image with predicted boxes around the detected objects. You can find the generated image at the following path: 🖼 **images/dif2.jpg**. The predicted objects and their corresponding scores are as follows: - Giraffe with a score of 0.999- Zebra with a score of 0.998- Zebra with a score of 0.998- Giraffe with a score of 0.969- Zebra with a score of 0.997. I hope this information is helpful to you.

**Execution Results #3:**
**1.** {'generated image with predicted box': '/images/dif2.jpg', 'predicted': [{'box': {'xmax': 760, 'xmin': 515, 'ymax': 437, 'ymin': 136}, 'label': 'giraffe', 'score': 0.9992127418518066}, {'box': {'xmax': 561, 'xmin': 510, 'ymax': 447, 'ymin': 357}, 'label': 'zebra', 'score': 0.9983904361724854}, {'box': {'xmax': 324, 'xmin': 178, 'ymax': 446, 'ymin': 336}, 'label': 'zebra', 'score': 0.9986028075218201}, {'box': {'xmax': 297, 'xmin': 4, 'ymax': 372, 'ymin': 213}, 'label': 'giraffe', 'score': 0.9694724082946777}]}

Figure 6: Case study on complex tasks (a).

**Query:** Describe this image example.jpg in as much detail as possible.

**Response:** The image you provided appears to show **a family of four dogs playing in the grass**. For this task, I ran multiple CV tasks, including image-to-text, image-classification, object-detection, visual-question-answering, and document-question-answering. I chose specific models for each task that were best suited to addressing your query, based on their training and performance characteristics.

For image-to-text, I chose the 🤗 **nlpconnect/vit-gpt2-image-captioning** model. This model generates an image caption with the given image. I applied this model to the input image and the developed output was: **a family of four dogs are playing in the grass** For image-classification, I selected the 🤗 **google/vit-base-patch16-224** model. This model is trained on natural images dataset and it can predict the label of the image output. I applied this model to the image and get the results showing the risk of each label. It shows the highest risk at "**Rhodesian ridgeback**" with a score of 93.8% For object-detection, I used 🤗 **facebook/detr-resnet-101 model**. This model is trained on COCO panoptic segmentation for object detection. I used this model to generate an image file with the predicted box. The generated image with predicted box is at: 🖼️ **/images/sd48.jpg** Regarding your request for image segmentation, I have completed the task using a pre-trained model called 🤗 **facebook/detr-resnet-50-panoptic**. The segmentation mask shows the different regions of the image that belong to different classes. In this case, the model has identified two main classes in the image, which are "**dog**", "**grass**" and "**tree**". The generated image with the segmentation mask can be found at 🖼️ **/images/t4s1.jpg** For visual question answering, I used 🤗 **dandelin/vilt-b32-finetuned-vqa** model. This model is a Vision-and-Language Transformer (ViLT) model, which is fine-tuned on VQAv2. It is suitable for visual-question-answering tasks, and is likely to provide accurate results for the given image. For the question "what are in the image?" It reply with a prediction **"dogs"** and the confidence is 84.9%.

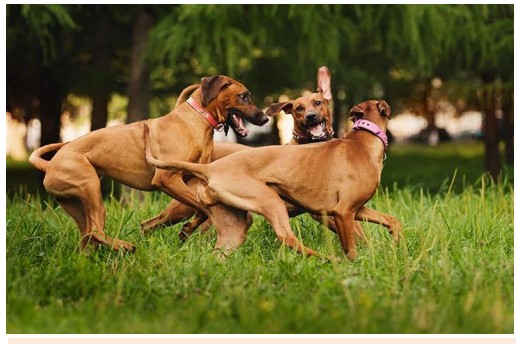

**Task Planning:**
**1.** {"id": "0", "dep": [-1], "task": "image-to-text", "args": {"image": "example.jpg" }},
**2.** {"id": "1", "dep": [-1], "task": "image-classification", "args": {"image": "example.jpg" }},
**3.** {"id": "2", "dep": [-1], "task": "object-detection", "args": {"image": "example.jpg" }},
**4.** {"id": "3", "dep": [-1], "task": "image-segmentation", "args": {"image": "example.jpg" }},
**5.** {"id": "4", "dep": [-1], "task": "visual-question-answering", "args": {"text": "describe this image in detail", "image": "example.jpg" }},

**Execution Results:**
**1.** {"generated_text": "a family of four dogs are playing in the grass "}
**2.** [{'score': 0.938717246055603, 'label': 'Rhodesian ridgeback'}, {'score': 0.01652258075773716, 'label': 'boxer'}, {'score': 0.006381669547408819, 'label': 'Great Dane'}, {'score': 0.006234415341168642, 'label': 'vizsla, Hungarian pointer'}, {'score': 0.005944834090769291, 'label': 'bull mastiff'}]
**3.** [{"generated image with predicted box": "/images/sd48.jpg","predicted":  [{"box": {"xmax": 463, "xmin": 373, "ymax": 267, "ymin": 199}, "label": "bus", "score": 0.9981155395507812}, {"box": {"xmax": 292, "xmin": 124, "ymax": 224, "ymin": 175}, "label": "airplane", "score": 0.9983609318733215}, {"box": {"xmax": 554, "xmin": 487, "ymax": 277, "ymin": 226}, "label": "person", "score": 0.9910836219787598}]}]
**4.** {"generated image with segmentation": "/images/t4s1.jpg", "predicted": [{"score": 0.989, "label": "grass"}, {"score": 0.999, "label": "dog"}, {"score": 0.999, "label": "tree"},{"score": 0.999, "label": "dog"}]
**5.** [{'answer': 'dogs', 'score': 0.8488452434539795}, {'answer': 'dog', 'score': 0.04168461635708809}]

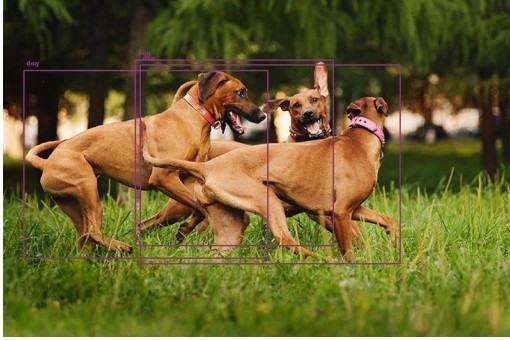

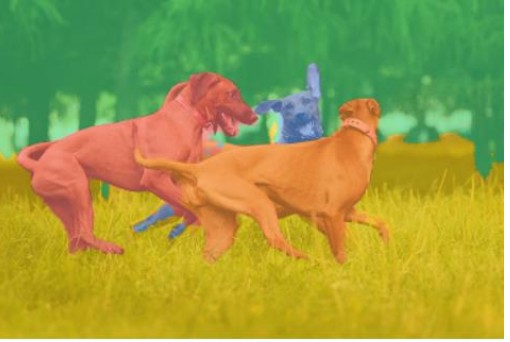

Figure 7: Case study on complex tasks (b).

**Query:** Can you tell me how many objects in the picture example1.jpg?

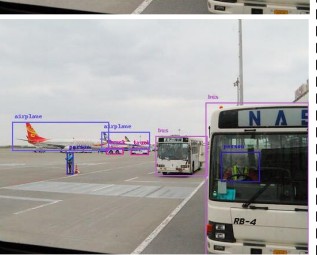

**Response:** There are **8 objects** in the picture. I used the model 🤗 **facebook/detr-resnet-101** for this task which is trained on a popular object detection dataset called COCO, and it has the highest number of downloads. It generated an image with predicted boxes that is available under the path: 🖼 **/images/79f2.jpg** . As for image captioning, I selected the model 🤗 **nlpconnect/vit-gpt2-image-captioning** which is more frequently used and works better in this case. The generated text from the image is '**a large jetliner sitting on top of an airport tarmac.**'

**Task Planning:**
**1:** {"task": "object-detection", "id": 0, dep: [-1], "args": {"image": "example1.jpg"}},
**2:** {"task": "image-to-text", "id": 1, dep: [-1], "args": {"image": "example1.jpg"}}

**Execution Results:**
**1.** [{"generated image with predicted box": "/images/79f2.jpg", "predicted": [{"box": {"xmax": 463, "xmin": 373, "ymax": 267, "ymin": 199}, "label": "bus", "score": 0.9981155395507812}, {"box": {"xmax": 292, "xmin": 124, "ymax": 224, "ymin": 175}, "label": "airplane", "score": 0.9983609318733215}, {"box": {"xmax": 554, "xmin": 487, "ymax": 277, "ymin": 226}, "label": "person", "score": 0.9910836219787598}, {"box": {"xmax": 230, "xmin": 219, "ymax": 264, "ymin": 227}, "label": "person", "score": 0.998579740524292}, {"box": {"xmax": 361, "xmin": 330, "ymax": 231, "ymin": 220}, "label": "truck", "score": 0.9066138863563538}, {"box": {"xmax": 639, "xmin": 460, "ymax": 420, "ymin": 142}, "label": "bus", "score": 0.9987366795539856}, {"box": {"xmax": 363, "xmin": 279, "ymax": 222, "ymin": 192}, "label": "airplane", "score": 0.9976897239685059}, {"box": {"xmax": 317, "xmin": 287, "ymax": 230, "ymin": 219}, "label": "truck", "score": 0.9339178800582886}]},
**2.** {"generated_text": "a large jetliner sitting on top of an airport tarmac "}

---

**Query:** In example2.jpg, what's the animal and what's it doing?

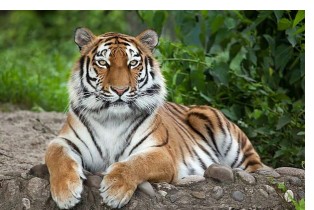

**Response:** Sure, **the animal in the image is a tiger and it's laying down.** I performed 3 tasks including image-to-text, image-classification, and visual-question-answering on the image you provided. To accomplish this, I selected models 🤗 **nlpconnect/vit-gpt2-image-captioning**, 🤗 **google/ vit-base-patch16-224** and 🤗 **dandelin/vilt-b32-finetuned-vqa** because they are the perfect models for this use case respectively. According to the results, I have 88% confidence that the animal is a tiger and also have a 77.8% confidence that it's laying down .The generated image with predicted box is stored at the path: 🖼 **/images/d59a.jpg**.

**Task Planning:**
**1:** {"task": "image-classification", "id": 0, dep: [-1], "args": {"image": "example2.jpg" }},
**2:** {"task": "visual-question-answering", "id": 1, dep: [-1], "args": {"text": "What is the animal doing?", "image": "example2.jpg" }}
**3:** {"task":"image-to-text", "id": 2, dep: [-1], "args": {"image": "example2.jpg" }}

**Execution Results:**
**1.** [{'score': 0.8858247399330139, 'label': 'tiger, Panthera tigris'}, {'score': 0.10940514504909515, 'label': 'tiger cat'}, {'score': 0.0006216467591002584, 'label': 'jaguar, panther, Panthera onca, Felis onca'}, {'score': 0.0004262701258994639, 'label': 'dhole, Cuon alpinus'}, {'score': 0.00030842673731967807, 'label': 'lion, king of beasts, Panthera leo'}]
**2.** [{'answer': 'sitting', 'score': 0.8294351696968079}, {'answer': 'laying down', 'score': 0.7387925386428833}, {'answer': 'resting', 'score': 0.625247061252594}, {'answer': 'laying', 'score': 0.6114301681518555}, {'answer': 'relaxing', 'score': 0.44740357995033264}]
**3.** {'generated_text': 'a brown and white striped zebra laying on a rock'}

Figure 8: Case study on complex tasks (c).

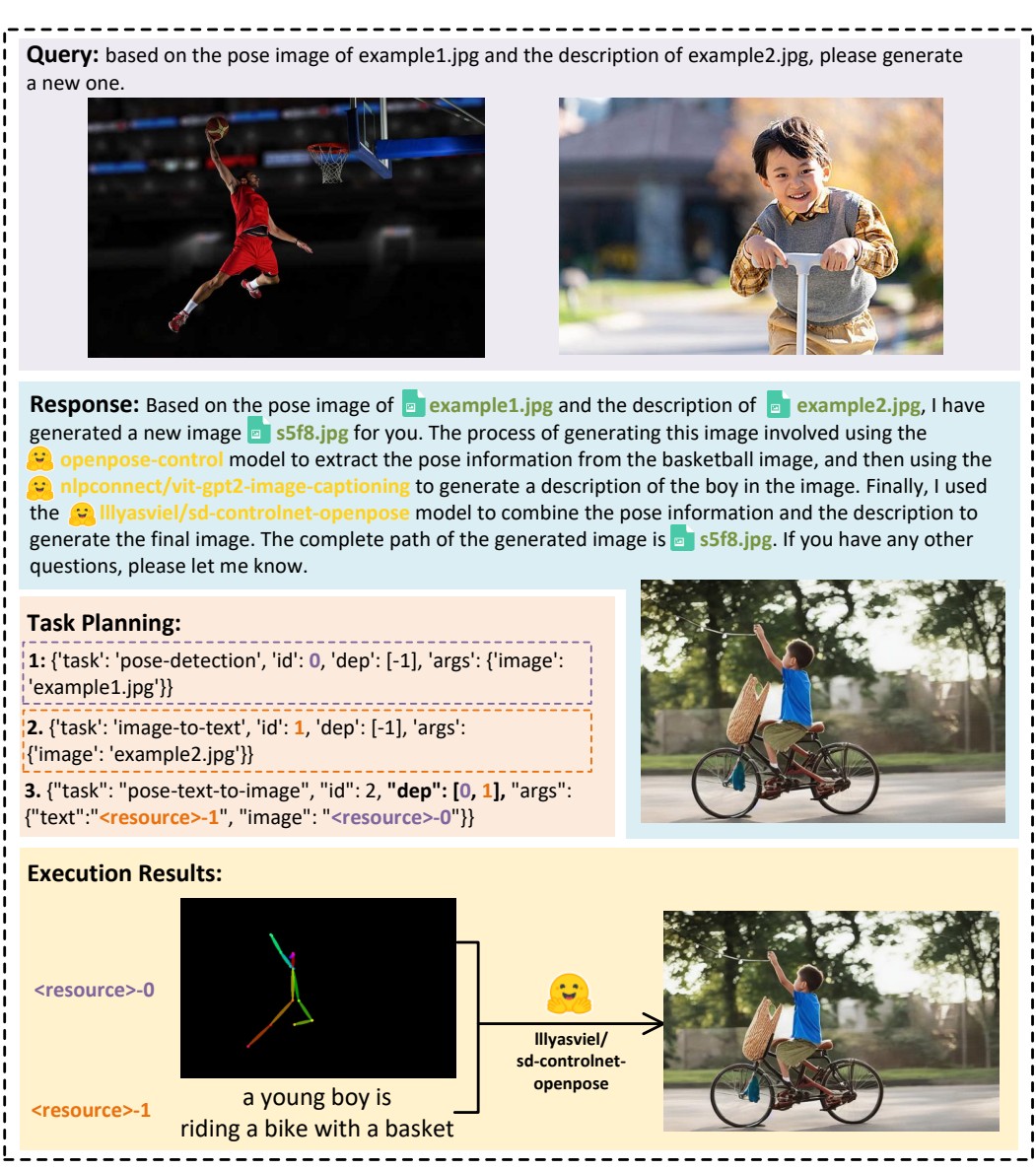

Figure 9: Qualitative analysis of multi-model cooperation with resource dependency.

**Query:** generate a video with the title "An astronaut is walking in space" and dub it.

**Response:** Sure. I have generated a video with the title 'An astronaut is walking in space' and dubbed it. The workflow I used is as follows: First, I used the text-to-video model ' 👧 **damo-vilab/text-to-video-ms-1.7b**' to generate a video from the text. The inference result is the generated video ' 🎥 **/videos/5696.mp4**'. Second, I used the text-to-speech model ' 👧 **facebook/tts_transformer-es-css10**' to generate an audio from the text. The inference result is the generated audio ' 🎵 **/audios/3f9e.flac** '. Finally, I combined the video and audio to generate the dubbed video. I hope this answers your question. Is there anything else I can help you with?

**Task Planning:**
**1:** {"task": "text-to-video", "id": 0, "dep": [-1], "args": {"text": "An astronaut is walking in space" }}
**2:** {"task": "text-to-speech", "id": 1, "dep": [-1], "args": {"text": "An astronaut is walking in space" }}

**Execution Results:**
**1.** {"generated video": "/videos/5696.mp4"}        **2.** {"generated audio": "/audios/3f9e.flac"}

---

**Query:** read the image example.jpg for me.        **INVOICE NO : 18301/102/T0305**

**Response:** Sure. I have read the image 🖼 **example.jpg** for you. The inference result is **'INVOICE NO : 18301/102/T0305'**. The model used for this task is 👧 **microsoft/trocr-base-printed**, which is best suited for the task of image-to-text as it is fine-tuned on the SROIE dataset which is specifically designed for this task. Then I have converted the text to audio, the generated audio is 🎵 **/audios/da5s.wav** 🔊 ⸾⸾|⸾ ⸾|⸾ ⸾|⸾ ⸾|⸾ and the model used for this task is 👧 **facebook/fastspeech2-en-ljspeech**, which is a FastSpeech 2 text-to-speech model which is suitable for the task of text-to-speech. Is there anything else I can help you with?

**Task Planning:**
**1:** {"task": "image-to-text", "id": 0, "dep": [-1], "args": {"image": "example.jpg"}}
**2:** {"task": "text-to-speech", "id": 1, "dep": [0], "args": {"text": "<resource-0>"}}

**Execution Results:**
**1.** {"generated text": "INVOICE NO : 18301/102/T0305"}    **2.** {"generated audio": "/audios/da5s.wav"}

Figure 10: Qualitative analysis of multi-model cooperation on video and audio modalities.

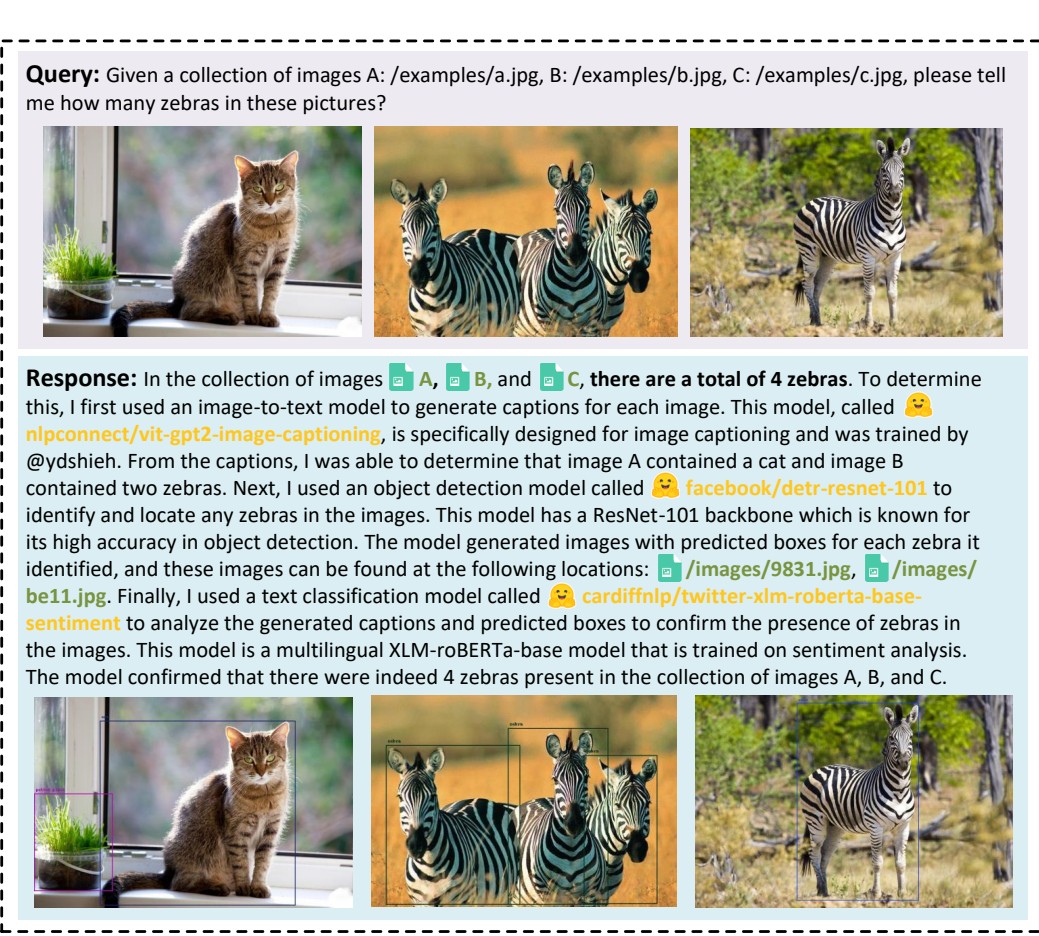

**Query:** Given a collection of images A: /examples/a.jpg, B: /examples/b.jpg, C: /examples/c.jpg, please tell me how many zebras in these pictures?

**Response:** In the collection of images 🖼️ **A,** 🖼️ **B,** and 🖼️ **C,** **there are a total of 4 zebras**. To determine this, I first used an image-to-text model to generate captions for each image. This model, called 🤗 **nlpconnect/vit-gpt2-image-captioning,** is specifically designed for image captioning and was trained by @ydshieh. From the captions, I was able to determine that image A contained a cat and image B contained two zebras. Next, I used an object detection model called 🤗 **facebook/detr-resnet-101** to identify and locate any zebras in the images. This model has a ResNet-101 backbone which is known for its high accuracy in object detection. The model generated images with predicted boxes for each zebra it identified, and these images can be found at the following locations: 🖼️ **/images/9831.jpg,** 🖼️ **/images/ be11.jpg.** Finally, I used a text classification model called 🤗 **cardiffnlp/twitter-xlm-roberta-base-sentiment** to analyze the generated captions and predicted boxes to confirm the presence of zebras in the images. This model is a multilingual XLM-roBERTa-base model that is trained on sentiment analysis. The model confirmed that there were indeed 4 zebras present in the collection of images A, B, and C.

Figure 11: Qualitative analysis of multi-model cooperation with multiple sources.

