# OpenReview forum: "HuggingGPT: Solving AI Tasks with ChatGPT and its Friends in Hugging Face"
_NeurIPS.cc/2023/Conference — NeurIPS 2023 poster_

### Official Review · Reviewer_FvPX · 2023-06-25

**Soundness:** 3 good
**Presentation:** 3 good
**Contribution:** 2 fair
**Rating:** 6
**Confidence:** 5

**Summary:**

The paper studies an interesting and important question, i.e., how to automate LLMs to call existing models for solving specific tasks. The authors propose a novel framework that contains the following steps: (1) task planning, (2) model selection, (3) task execution, and (4) response generation. The experimental results well support the claim. By leveraging ChatGPT and abundant AI models Hugging Face, the proposed method is able to cover numerous sophisticated AI tasks in different modalities and domains and achieve impressive results in many tasks.

**Strengths:**

+ The paper is well-written and easy to follow.

+ A novel idea proposed and well-supported experiments conducted.

**Weaknesses:**

- The method heavily relies on the existence of SOTA LLMs (e.g., ChatGPT), which may involve some practical issues (e.g., unaffordable API costs). It remains under-explored whether open-source LLMs (e.g., vicuna) can be leveraged for the framework (or how to adapt vicuna for better task planning).

**Questions:**

For the model selection part, what if the number of models hosted on Huggingface becomes extremely large? How to ensure that the model selects the most appropriate one from them?

---

> ### Author Rebuttal · Authors · 2023-08-10
>
> Thanks for your valuable comments. Below are our responses to your concerns:
>
> ***Q1: Whether open-source LLMs can be leveraged for the framework?***
>
> Yes. The open-source LLMs are also suitable for our framework. In our experiments (please see Table 3, 4, 5), we also deploy open-source LLMs for evaluation. Currently, although open-source LLMs are still weaker than ChatGPT and GPT-4, we believe these gaps can be reduced with the development of open-source LLMs, for example, LLaMA-2 [1]. Besides, by collecting high-quality user instruction data, we can fine-tune these open-source LLMs to better utilize these AI models from ML communities for solving AI tasks. Some recent works [2] also indicate the availability of this strategy.
>
> ***Q2: What if the number of models hosted on Huggingface becomes extremely large? How to ensure that the model selects the most appropriate one from them?***
>
> Thanks for your question. Generally, it could possibly affect the accuracy of the selected model and bring additional costs due to longer contexts (e.g., model descriptions), when the number of models hosted on Hugging Face becomes extremely large. To handle this problem, in our framework, we currently design a filtering strategy (please see section 3.2) to guarantee the stability of the system: 1) for the current task, we first retain the models whose task type can match it; 2) then we filter out the top-K candidate models based on the number of their downloads; 3) finally, we enable LLMs to select the most suitable model from these candidate models by considering the user request. We will make a more clear description of this part in the final version and continue to investigate this problem in the future.
>
> Besides, our strategy is just one of these solutions and we believe there should have more optimal strategies for this problem, which can be studied in the future. For example: 1) we could construct a hierarchy graph of all models based on the model cards on Hugging Face, to enable better selection; 2) by collecting many user requests which include model(tool)-use, we could optimize public LLMs (e.g., LLaMA-2) to enable it to make a better strategy for model selection. We will also continue to investigate this direction to search for better strategies.
>
> [1] LLaMA 2: Open Foundation and Fine-Tuned Chat Models
>
> [2] ToolLLM: Facilitating Large Language Models to Master 16000+ Real-world APIs.

---

### Official Review · Reviewer_WQcy · 2023-07-06

**Soundness:** 3 good
**Presentation:** 4 excellent
**Contribution:** 4 excellent
**Rating:** 7
**Confidence:** 4

**Summary:**

This paper presents a pipeline to manipulate many autonomous agents (mainly open-sourced models in  Hugging Face) . Tother with these models could solve NLP, CV, audio and Video tasks,  the resulted HuugingGPT could could complicated multi-modal tasks that might be decomposed a sequence  of atomic  tasks or a graph.


**Strengths:**

- The phylosophy is interesting and insightful. I like the idea very much.
- This seems be a promising direction to solve multi-modal tasks using HuggingGPT.


**Weaknesses:**

- The whole pipeline seems to provide a series of prompts to solve some combined tasks. The method is not scientific from a traditional point of view.
- The evalaution protocol seems not mature.  For example, there is not evidence to check whether the evalution makes sense or not.


**Questions:**

- It seesm the authors manually filter top-K models based on the number of downloads. What is the exact number of models in each task? and how do we select the best model among the top-K models? based on which criterion?


- It seems that it uses different evaluation metrics in table 3,4, and 5? Is there any explaination?  Can we use some straightforward metrics like  task success rate in Table 4 and 5?

- We do not know quality of the GPT-4 anotated data during evaluation. Is it possiable to invite experts to manually check some of them.  Don't you think reporting results in human-annotated and GPT-4 anotated data separately could be a better idea.


- Will these evaluation examples  and evaluation scripts be open-sourced?

**Limitations:**

- Evalution is weak

---

> ### Author Rebuttal · Authors · 2023-08-10
>
> Thanks for your valuable comments. Below are our responses to your concerns:
>
> ***Q1: The method is not scientific from a traditional point of view.***
>
> Thanks for your question. The success of our framework benefits from the advent of powerful LLMs (i.e., ChatGPT or GPT-4). Therefore, we adopt a series of prompts as a workflow to implement each step and establish the connection between each stage. By introducing these well-designed prompts,  our framework exhibits powerful capabilities in autonomously analyzing user requests and selecting suitable models for solving tasks. This method also verifies the feasibility of using LLMs as the controller to manage AI models. With the advent of more powerful open-source LLMs (e.g., LLaMA-2), we will introduce more advanced learning strategies into our framework.
>
> ***Q2: What is the exact number of models in each task? How do we select the best model? Based on which criteria?***
>
> Thanks for your question. Here, we filter out the Top-K models based on the number of downloads (lines 167-168). Due to the limitation of prompt length, we set the number of candidate models for each task as 20 (i.e., the value of K). And for each task, its original number of models before filtering is usually varying (the average number is nearly 60 but some tasks only have few models). After that, we do not set up any fixed criteria but let LLMs automatically select the best model from the Top-k models by providing these model descriptions, which include its downloads, domain, task type and etc. We think that LLMs could select the best suitable models after comprehensively considering these information. We will add these details to our final version.
>
> ***Q3: Explanations about different metrics in Table 3, 4, and 5? Can we use some straightforward metrics like task success rate in Table 4 and 5?***
>
> Generally, we think any user request can be converted as a directed acyclic graph (DAG). However, directly evaluating DAG is quite difficult. Therefore, based on the complexity, we divide user requests into three types (i.e., single task, sequential task, and graph task). We first use Recall, Precision, and F1-score as the uniform metrics. But for complex tasks (sequential or graph tasks), only these metrics cannot reflect the planning capability in task dependency. Hence, we additionally use [edit distance](https://en.wikipedia.org/wiki/Edit_distance) to estimate the matching degree in sequential tasks. And for the graph task, since it is very difficult to measure the matching degree of graph data, we currently adopt GPT-4 as a subjective metric to measure its performance.
>
> And thanks for your suggestions. We have conducted human evaluations to better measure the planning capability and final generation of LLMs in our framework. Here, we design passing rate, rationality and success rate as the subjective metrics to evaluate different stages:
>
> - Passing rate: Check LLMs whether can understand instructions and execute in the current stage.
> - Rationality: Check whether the generated task sequence / selected tools are rational when compared with user requests.
> - Success rate: Check the final results whether satisfy the user request.
>
> We invite three human experts to annotate the provided data based on our designed metrics, and then calculate the average value as the final score. The results of human evaluation are reported in Table 2. These results can further validate the effectiveness of our proposed framework. We will continue to design more reasonable metrics for evaluation and add these results into our final version.
>
> | **LLM** | **Task Planning** | | **Model Selection** | | **Response Generation**  |
> |-|-|-|-|-|-|
> |  | **Passing Rate** ↑  | **Rationality** ↑  | **Passing Rate** ↑  | **Rationality** ↑  | **Success Rate**↑     |
> | Alpaca-13b   | 51.04      | 40.26    | -   | -     | 6.92    |
> | Vicuna-13b   | 79.41      | 66.15  | -  | -  | 15.64    |
> | GPT-3.5      | 99.42      | 91.03 | 93.89   | 84.29    | 63.08     |
>
> *Table 2. Human Evaluation. We design passing rate, rationality and success rate as the subjective metrics and invite human experts to evaluate them. (Same as Table 2 in Rebuttal PDF.)*
>
> ***Q4: Reporting results in human-annotated and GPT-4 annotated data separately.***
>
> Thanks for your suggestion. Actually, we have separated results into human-annotated and GPT-4 annotated data.  Table 3, 4, and 5 reported results on GPT-annotated data (in single, sequential, and graph tasks). We will make a more clear description of this part in the final version. Furthermore, we also invite human experts to conduct human evaluations for our experiments to further verify our results. The results are shown in Table 2 (please see Rebuttal PDF).
>
> ***Q5: Will these evaluation examples and evaluation scripts be open-sourced?***
>
> Yes. Currently, we are inviting the labeling team to conduct human evaluation and help us to iteratively clean up the whole dataset. We will release all evaluation examples and the evaluation scripts in the final version.

---

### Official Review · Reviewer_sGd8 · 2023-07-06

**Soundness:** 4 excellent
**Presentation:** 4 excellent
**Contribution:** 4 excellent
**Rating:** 9
**Confidence:** 4

**Summary:**

The authors propose HuggingGPT , a collaborative system for solving AI tasks, which is composed of a large language model (LLM) and numerous expert models from ML communities. They provide methods for each of the four stages involved in HuggingGPT's workflow: task planning, model selection, task execution, and response generation.

**Strengths:**

Impactful and well written paper
- simple strategy for handing resource dependencies for executing tasks
- compelling idea of chaining of expert models to provide a tool to decompose a task into sub-tasks and identify the appropriate expert models to solve these sub-tasks
- reasonable format for inputting task request along with examples from the user
- robust evaluation  - human evaluation along with automated evaluation

**Weaknesses:**

I don't see any weaknesses in the experiments, evaluation or novelty of this paper.

**Questions:**

No questions.

**Limitations:**

Yes, the authors list the limitations of their work as a separate section.

---

> ### Author Rebuttal · Authors · 2023-08-10
>
> We sincerely thank you for your positive comments. We will continue to refine our paper and devote more effort to the subsequent works to facilitate the community to better understand and explore this new research direction.

---

### Official Review · Reviewer_FNMr · 2023-07-07

**Soundness:** 3 good
**Presentation:** 4 excellent
**Contribution:** 4 excellent
**Rating:** 6
**Confidence:** 4

**Summary:**

This paper considers large language models (LLMs) like ChatGPT as a controller and presents a new framework called HuggingGPT, which connects various AI models in the existing ML community (i.e., HuggingFace). Specifically, HuggingGPT consists of four steps including task planning, model selection, task execution, and response generation. By leveraging the strong capability of LLMs and numerous AI models in different modalities, HuggingGPT can solve sophisticated AI tasks and achieve promising results.

**Strengths:**

1) The idea to connect LLMs with rapidly developing ML communities like HuggingFace is novel. It largely extends the applicability of LLMs to solve multi-modality AI tasks by fully utilizing the existing powerful models in HuggingFace.
2) Each step of HuggingGPT is well designed from Figure 2. The whole paradigm of HuggingGPT is neat and effective.
3) This paper is well written and easy to follow.


**Weaknesses:**

1. In Section 3.1, the demonstration examples may have an important impact on the parsing performance. The authors should provide more details such as the number of demonstration examples and the method to select these examples. The demonstration case provided in Table 1 is somewhat confusing for me because it only involves the tasks about images and texts. Can these demonstration examples benefit the parsing of tasks in other modalities like audio? I also wonder whether the demonstration examples for each user request are the same.

2. In Section 3.2, the authors propose a model selection strategy based on in-context alignments and the number of downloads. But in my view, the contribution of this module is questionable. Since the performance gap between different models for each task may be significantly large due to model scales (e.g., GPT-2 and LLaMA for text generation), it’s nearly impossible to select weaker models to dealing with the corresponding task. Thus, I’m curious about the performance if we directly use the best model for each task. There are also no empirical results to show the necessity of model selection.

3. The experimental result is somewhat weak in terms of the following points:

(1) The authors only conduct the empirical analysis on task planning. Other modules should be also tested individually.

(2) The authors only use automatic evaluation metrics to measure the model performance. However, even GPT-4 score may have potential biases in the evaluation of generated texts. It’s better to involve human evaluation to make experimental results stronger.

**Questions:**

I have included my questions in the weaknesses part.

**Limitations:**

The authors have adequately addressed the limitations.

---

> ### Author Rebuttal · Authors · 2023-08-10
>
> Thanks for your valuable comments. Below are our responses to your concerns:
>
> ***Q1: More details about demonstration examples.***
>
> The number of demonstration examples is set as 3 in our default settings. Here, we select demonstration examples that contain more tasks and complex task dependencies, and simultaneously use fewer tokens in prompts. The target of using these demonstration examples is to expect LLMs to understand how to parse the task queue. Therefore, even if these examples only involve text and image tasks, our framework will still make correct predictions on audio tasks. And currently, the demonstration examples are set as the same for the user.
>
> Furthermore, we also conduct different experiments to investigate the effects of demonstration examples, from the perspectives of variance and the number of demonstrations. Here, we deploy different numbers of task types involved in the demonstrations to measure the variance in demonstration examples. The results can be found in Figure 1 (please see Rebuttal PDF) and Table 1. We found that increasing variance or the number of demonstrations can both improve the planning capability of LLMs. In the future, we will continue to investigate more elements to explore their effects on our framework.
>
> | **Demo Variance** | **LLM** | **Single Task**    |  | **Sequential Task**   |  | **Graph Task** |  |
> |:------|:----|:-----|:---------|:-----|:--------:|----|-----|
> |                                       |         | **Acc** ↑ | **F1** ↑  | **ED (%)**  ↓  | **F1** ↑  | **F1**    ↑    |
> | 2                                     | GPT-3.5  | 43.31      | 48.29     | 71.27        | 32.15     | 43.42         |
> | 2                                     | GPT-4    | 65.59      | 67.08     | 47.17        | 55.13     | 53.96         |
> | 6                                     | GPT-3.5  | 51.31      | 51.81     | 60.81        | 43.19     | 58.51         |
> | 6                                     | GPT-4    | 66.83      | 68.14     | 42.20        | 58.18     | 64.34         |
> | 10                                    | GPT-3.5  | 52.83      | 53.70     | 56.52        | 47.03     | 64.24         |
> | 10                                    | GPT-4    | 67.52      | 71.05     | 39.32        | 60.80     | 66.90         |
>
> *Table 1. Evaluation of task planning in terms of the diversity of demonstrations. (Same as Table 1 in Rebuttal PDF.)*
>
> ***Q2: Can we directly use the best model for each task? The necessity of model selection?***
>
> Yes. The motivation of model selection is to answer which model is suitable for our task, and expect us to determine a selection strategy. Here, directly selecting the best model can be considered as an available strategy. However, "the best model" is difficult to define, since it may involve too many elements. For example, which datasets, metrics and domains to measure the best model. And when involving some domain-specific requests (as shown in Table 3 of Rebuttal PDF), e.g., creating anime-style images, generating protein sequences, etc., it is difficult to cope with such scenarios by directly selecting the model with the best performance.
>
> Therefore, we expect to fully utilize the provided information (e.g., model structure, training data, applicable domain, performance, download, and etc) in the model card to more comprehensive support for model selection. Besides, as mentioned in our comments (Q1) to all reviewers, keeping model selection individually can also enable better efficiency, flexibility, and extensibility. In the future, we will continue to explore better strategies for model selection.
>
> ***Q3: Other modules should also be tested individually.***
>
> Thank you for your suggestions. In the initial version, we mainly evaluate our method on task planning since this module is very important to determine the subsequent results. Following your suggestions, we have added more experiments to evaluate demonstrations (please see Q1), and human evaluation on model selection and response generation (please see Q4). We found that the capability of LLMs could also affect the results of model selection, so how to optimize model selection is also very important. In the future, we will conduct more experiments to investigate more elements in our framework.
>
>
> ***Q4: Involve human evaluation.***
>
> Thanks for your suggestions. We further collected 130 diversity requests and have conducted human evaluations to better measure the planning capability and final generation of LLMs in our framework. Here, we design passing rate, rationality and success rate to evaluate different stages:
>
> - Passing rate: Check LLMs whether can understand instructions and execute in the current stage.
> - Rationality: Check whether the generated task sequence / selected tools are rational when compared with user requests.
> - Success rate: Check the final results whether satisfy the user request.
>
> We invite three human experts to annotate the provided data based on our designed metrics, and then calculate the average value as the final score. The results of human evaluation are reported in Table 2 (please see Rebuttal PDF). These results can further validate the effectiveness of our proposed framework. We will continue to design more reasonable metrics for evaluation and add these results into our final version.
>
>
> | **LLM** | **Task Planning** | | **Model Selection** | | **Response Generation**  |
> |-|-|-|-|-|-|
> |              | **Passing Rate** ↑  | **Rationality** ↑  | **Passing Rate** ↑  | **Rationality** ↑  | **Success Rate**↑     |
> | Alpaca-13b   | 51.04      | 40.26    | -   | -     | 6.92    |
> | Vicuna-13b   | 79.41      | 66.15             | -  | -                | 15.64    |
> | GPT-3.5      | 99.42      | 91.03             | 93.89       | 84.29    | 63.08     |
>
> *Table 2. Human Evaluation. We design passing rate, rationality and success rate as the subjective metrics and invite human experts to evaluate them. (Same as Table 2 in Rebuttal PDF.)*

---

> > ### Comment · Reviewer_FNMr · 2023-08-18
> > **Response to Rebuttal**
> >
> > Thanks for your rebuttal. The additional experimental results in Q1 and Q2 are meaningful for me. As for Q4, I expect that more details about human evaluation should be added to the final version of the paper, such as the modality / task distribution of 130 requests, the human annotation instructions, and the agreement among 3 human experts.

---

> > > ### Author Response · Authors · 2023-08-19
> > > **Thanks for Your Response**
> > >
> > > Dear Reviewer,
> > >
> > > Thank you for taking the time to review our rebuttal and for providing constructive feedback. We are pleased to hear that the additional experimental results in Q1 and Q2 have addressed your concerns. Based on your comments, we provide more details regarding the human evaluation and will add them to our final version. Please see as follows:
> > >
> > > **Modality and Task Distribution**
> > >
> > > To conduct a more comprehensive evaluation, we collected a dataset for human evaluation that covers as many modalities and tasks as possible. Below are the statistics on modality and task distribution.
> > >
> > > The statistics for modality distribution are as follows:
> > >
> > > |Modality| Count |
> > > |-|-|
> > > |Text | 46 |
> > > |Text + Image |58|
> > > |Text + Video |7|
> > > |Text + Audio |1|
> > > |Text + Image + Video |3|
> > > |Text + Image + Audio |2|
> > > |Text + Video + Audio |8|
> > > |Text + Image + Audio + Video|5|
> > >
> > >
> > > We also conducted a statistical analysis of the detailed task types in the task graph. The statistics are as follows:
> > >
> > > |Task| Count |
> > > |-|-|
> > > |image-to-text|52 |
> > > |text-to-image|38|
> > > |visual-question-answering|30|
> > > |text-to-video|23|
> > > |conversational|20|
> > > |text-to-speech|20|
> > > |object-detection|13|
> > > |text2text-generation |13|
> > > |image-to-image|9|
> > > |question-answering |6|
> > > |text-generation|4|
> > > |audio-to-audio|4|
> > > |image-classification| 3|
> > > |openpose-control |3|
> > > |openpose-text-to-image| 3|
> > > |canny-control|2|
> > > |canny-text-to-image|2|
> > > |automatic-speech-recognition|2|
> > > |image-segmentation|1|
> > > |seg-text-to-image |1|
> > > |translation|1|
> > > |audio-classification|1|
> > > |token-classification|1|
> > > |mlsd-control| 1|
> > > |mlsd-text-to-image|1|
> > > |summarization|1|
> > >
> > > Based on the modality and task distribution, we observe that our collected dataset for human evaluation covers text, image, audio, and video modalities as well as 27 tasks. These distributions suggest that our dataset for human evaluation is capable of good generalization, and can reflect the ability of HuggingGPT when processing real user requests.
> > >
> > > **Human Annotation Instructions**
> > >
> > > We invited three human experts to evaluate the outputs of the large language models, and formulated evaluation guidelines for the task planning, model selection, and response generation stages.
> > >
> > > In the task planning and model selection stages, we asked the experts to evaluate the passing rate and rationality of the outputs of the large language models:
> > >
> > > + Task planning or model selection is passed when and only when the task graph or the selected model is executable. We define a task graph as executable in three ways: the output is parsable, the argument types of the tasks are correct, and the interconnected tasks match in terms of resource dependencies. For model selection to be considered executable, the output must be parsable and the chosen model must be valid.
> > > + The rationality of task planning or model selection is a stricter metric that requires validation of correctness beyond just execution passes. Task planning is considered rational if the planned task graph is executable and has the potential to address the user's request. Likewise, model selection is considered rational if the chosen model is the most suitable among the candidate models to meet the user intent and can effectively complete the planned task.
> > >
> > > In the response generation stage, we asked human experts to evaluate whether the responses generated by HuggingGPT can eventually solve the user requests. Besides, if some media resources are generated based on user requests, we also require the experts to determine whether the generated media content satisfies the user requirements.
> > >
> > > Moreover, to make a fair evaluation, we anonymized the names of the large language models to prevent any biases based on prior experience.
> > >
> > > **Agreement among Human Experts**
> > >
> > > Thank you for your suggestion. To measure the agreement among 3 human experts in their evaluations, we designed three metrics which are as follows:
> > >
> > > + **Percent Agreement**: This represents the percentage of items where all raters agree.
> > > + **Average Pairwise Agreement**: This metric computes the agreement between pairs of raters and averages them. For three annotators, we assess agreement for pairs (1, 2), (1, 3), and (2, 3) and average the results.
> > > + **Fleiss' Kappa**: This measures the reliability of agreement among raters for categorical ratings. It extends Cohen's Kappa to multiple raters. Values range from -1 to 1: closer to 1 means stronger agreement, 0 indicates random agreement, and negative values indicate less than chance agreement.
> > >
> > > |LLM|Percent Agreement|Average Pairwise Agreement| Fleiss' Kappa|
> > > |:-|:-:|:-:|:-:|
> > > |Alpaca-13b|82.31%|88.21% |0.78|
> > > |Vicuna-13b|81.53% |87.69%|0.83|
> > > |GPT-3.5|77.69%|85.13%|0.69|
> > >
> > > We observe a high level of consistency in the human evaluations across the three models. Their evaluations align closely with each other, leading to moderate to excellent agreement metrics. This suggests that the evaluations are reliable. All of these details will be included in our final version.

---

### Official Review · Reviewer_hS4Z · 2023-07-07

**Soundness:** 2 fair
**Presentation:** 3 good
**Contribution:** 2 fair
**Rating:** 3
**Confidence:** 4

**Summary:**

This paper presents a framework that uses LLM as controller over modularized and specialized task models to plan and execute a complex task. The approach is to prompt LLM to decompose a given task command into a execution DAG, and for each step, parse model specifications (as metadata expressed in HuggingFace model cards) and select according modularized models for execution, and finally summarize a response to give it to the user. The idea is on the line with recent works on LLM-based planning and tool using.

**Strengths:**

I think the idea is rather novel. It aligns with recent works on using LLM as a central component to query more specialized models to complete a complex task. This paper proposes to exploit the vastly available models hosted on Hugging Face in a combinatorial way. If things work out well, it could have a substantial application impact.

**Weaknesses:**

The biggest selling point of this paper, as is repetitively mentioned in the paper itself, is the planning. But compared to recent works, the planning strategy in this paper is actually rather simple. Existing works on planning often involves an iterative process where LLM plans, executes, and observes, and improves (for instance, Reflexion, AdaPlanner, Self-Refine, etc). In this paper, it's just plan and execute. So, on the planning part, I do not see any contribution. Maybe, compared to the planning, a bigger contribution of this paper is the task decomposition.

Experiments are very limited.
- Data scale is quite small (46 trace annotations).
- No planning baseline, no ablation study, and no insight about model interplay.
- Comparisons are all on different LLMs; this has little to do with the claimed contribution on planning.

A simple baseline can be directly using specialized models. I think even if in some cases the proposed HuggingGPT does not outperform, it still gives reader a good picture about the pros and cons. An immediate ablation study I can think of is why not merge the model selection and task planning into one step, or what is the impact of model selection, especially given the large number of models on Hugging Face. I see no clue in this paper.

It is not clear how the proposed approach rely on the few-shot demonstration. Prior works on planning mostly rely on few-shot in-context prompting. It seems this paper is also on this technical line. But such dependency is also a limitation in the general use case. It's also no clear about the variance of the few-shot prompt used, e.g., whether the few-shot examples are fixed, and what about their diversity.

**Questions:**

Please see my general comments.

**Limitations:**

Please see my general comments.

--------------
**This section summarizes my concern based on rebuttal.**

There is a very simple baseline this paper needs to compare with. That is a simple one-pass reasoning by LLM without any implicit or explicit planning or model selection. The author so far refuses to do such ablation, therefore I am not convinced that the approach is effective. The proposed framework is indeed interesting, but an interesting thought remains a thought without proven effectiveness.

---

> ### Author Rebuttal · Authors · 2023-08-10
>
> Thanks for your valuable comments. Below are our responses to your concerns:
>
> ***Q1: The biggest selling point of this paper is planning but the planning strategy in this paper is actually rather simple.***
>
> Thanks for your question. Below are our answers:
>
> - First, we want to highlight that while planning is a crucial stage in our framework, the biggest contribution of our paper is to present how to utilize LLMs to coordinate multiple AI models from ML communities (e.g., Hugging Face) for solving AI tasks. And to the best of our knowledge, this is the first work to build connections between LLMs and AI models from ML communities.
> - Second, existing works (e.g., Reflexion, AdaPlanner, Self-Refine) for planning are mainly designed for solving language tasks (e.g., reasoning, coding, and etc). But the planning in our framework is to use LLMs for analyzing user requests and thus connecting external models for solving AI tasks. To fulfill this target, we introduce task decomposition with well-designed prompts plus demonstrations to instruct LLMs to conduct task planning. Therefore, essentially, our planning strategy is well designed and not simple, and the goal of our planning is quite different from existing planning works.
>
> ***Q2: Experiments are limited.***
>
> Thanks for your question. Below are our answers:
>
> - **Data scale is quite small**: Our dataset is composed of GPT-annotated (3,497 cases) and human-annotated (46 cases) data (please see Appendix A.2). As aforementioned, our work is the first attempt to connect LLMs with the ML community. As a result, there are no existing datasets for reference and we need to construct datasets by ourselves, which is very difficulty. Until now, only a few works (e.g., Gollria [1], ToolLLM [2]) released some datasets, but all of them are constructed by LLMs. However, we will try our best to build more high-quality data to further facilitate the investigation of this direction.
>
> - **No Planning baseline and ablation study**: As aforementioned, the motivation of our paper is to present how to connect various AI models by using LLMs, not just for designing a planning strategy. And most of the existing planning works are for solving language tasks, while ours is to use LLMs to conduct task planning and thus invoke other models for solving tasks. We also give a discussion about planning in Appendix B. Besides, we have added more ablation studies (Please see Rebuttal PDF).
>
> - **Comparisons are on different LLMs, not for planning**: The experiments on different LLMs are used to highlight the importance of planning in our framework, and planning could also be a critical metric to reflect the capability of LLMs. Besides, we also conduct a discussion (Appendix B) with other works (e.g., AutoGPT) in different dimensions, including planning.
>
> Besides, we also add more experiments, ablation studies, and human evaluations about other modules (Please see Rebuttal PDF).
>
> [1] Gorilla: large language model connected with massive apis
>
> [2] ToolLLM: Facilitating Large Language Models to Master 16000+ Real-world APIs
>
>
> ***Q3: Why not merge the model selection and task planning into one step?***
>
> Thanks for your question. We think it is necessary to keep the model selection as an individual step and give these reasons:
>
> 1. **Motivation**: First, task planning helps us to analyze user request and decompose it into a task sequence, while model selection is to answer which model is suitable for this task. As a result, the target of each stage is different and both of them play very important roles in our framework.
>
> 2. **Efficiency**: Sometimes, the number of models from ML communities could be massive. Since we need to use model descriptions to call the most suitable model, selecting models will bring more token consumption in prompts. As a result, combining task planning and model selection will cost more tokens within one query, and separating these two steps will be more friendly for LLM computations.
>
> 3. **Flexibility and Extensibility**: Furthermore, keeping task planning and model selection as two parts can provide better flexibility and extensibility: 1) the model library is usually dynamic since developers will continually add the latest models into the community. Therefore, we expect that the model selection is independent of task planning so that it is easy for us to update the model library; 2) some models could include more detailed granularity (e.g., specific domains). These settings necessitate the development of more flexible selection strategies.
>
> Therefore, we prefer to keep model selection and task planning individually to help us to develop each stage. We will also add these descriptions to our final version to better explain the design of our framework.
>
> ***Q4: How does the proposed method rely on the few-shot demonstration? And the variance of the few-shot prompt used?***
>
> Thanks for your question. We utilize the few-shot demonstration to instruct LLMs to understand user intent, task dependency, and how to parse task sequence. By introducing few-shot demonstrations, LLMs is able to produce better results by following our instructions. To further investigate the effect, we respectively conduct experiments to investigate the effect of the number of demonstration examples (0-5). The results are reported in Figure 1 (Please see Rebuttal PDF). We can find that the planning capabilities of LLMs can obtain significant improvements by introducing few-shot demonstrations.
>
> Besides, we also conduct experiments to investigate the variance of demonstration examples. Here, we test different numbers of task types involved in the demonstrations to measure the variance. The results can be found in Table 1 (please see Rebuttal PDF). We observe that increasing the variance of demonstration could also improve the planning capability of LLMs.

---

> > ### Comment · Reviewer_hS4Z · 2023-08-18
> > **Added materials are great bonus to the paper but still not to the point.**
> >
> > Thank you for the discussion and making the experiment results more complete. I agree with the design considerations.
> >
> > However, my biggest concern is still on the experiment part, especially on the planning. The paper and rebuttal material only offer verbal content about why planning could be useful. But I think the most important part is to show that **if it is indeed useful**. One very simple and easy-to-come-with experiment is comparing against solving some sequential/graph language-only tasks without any planning. Just throw these tasks to GPT-3.5 and compare the output with your model with planning.
> >
> > Unless I entirely missed something on this ablation in the paper, I lean on rejection. The idea is good, but it lacks evidence to support it.
> >
> >
> > ----
> >
> > Some minor points: there is also a lack of references to prior works on task decomposition which has been studied extensively in the field. The application scenario is different, but the concept remains the same.

---

> > > ### Author Response · Authors · 2023-08-19
> > > **( 1/2 ) Response to Comments**
> > >
> > > Thank you for taking the time to review our rebuttal. While we respectfully disagree with your opinions on our experimental part, we try our best to answer your question and address your concerns:
> > >
> > > + First, as mentioned in our previous comments, HuggingGPT is an integrated framework, which includes task planning, model selection, task execution, and response generation. Directly removing any stage in HuggingGPT would undermine its integrity and make it collapse. And benefiting from such a framework, HuggingGPT is able to formulate such a concept: connecting LLMs with AI models to autonomously address complex AI tasks from any modalities and any domains. That is the biggest contribution. And the experiments in our paper are used to highlight the importance of planning capability in such a framework, not to discuss how to formulate a better planning strategy. In our limitations (Please see lines 301-304), we also mention that we need to explore how to improve LLM capability in task planning in the future.
> > >
> > > + Secondly, the reviewer argues how to prove planning is indeed useful. As aforementioned, HuggingGPT is an integrated framework, and removing any step will make it collapse. For example, task planning should not only consider parsing the task sequence from user requests, but also guarantee the parsed task sequence can be understood by the model selection stage. Therefore, the target of our paper is to verify whether HuggingGPT can succeed. As a result, all experiments in our paper indicate that our framework can achieve promising results and the design of such a framework is **indeed useful** to connect LLMs with AI models from the ML community. Please note that this is the first successful attempt to validate the availability of this problem. As a result, these conclusions prove that each stage in HuggingGPT is critical to fulfilling the final target. And then, we conduct experiments to further reveal which stage in HuggingGPT is more important and can be improved, but not to discuss which planning strategy is optimal. We expect the reviewer can understand this point.

---

> > > > ### Author Response · Authors · 2023-08-19
> > > > **( 2/2 ) Response to Comments**
> > > >
> > > > + To address your concerns, we also conduct experiments to discuss ChatGPT and HuggingGPT. Before presenting the experiments, we want to highlight that ChatGPT is also regarded as a model (or tool) at the model selection stage of our framework, for handling generative tasks. In other words, our framework encompasses the capabilities of the original ChatGPT. Here we select two examples to compare the outputs of each setting: 1) sequential / graph language-only tasks; 2) tasks involving multi-modality. And the results are shown in Table below. For language-only tasks, ChatGPT will utilize only its language ability to produce answers. When involving more expert knowledge, HuggingGPT can invoke more expert models (e.g., BioGPT) to address. When switching to multi-modality tasks, ChatGPT will directly refuse to answer the user, whereas HuggingGPT is equipped to process multi-modality information. These results also indicate the advantage of HuggingGPT in solving AI tasks from any domain and any modality.
> > > >
> > > > + Moreover, when you require us to make a comparison in language-only tasks also underscores the limitations of existing planning or LLMs tailored exclusively for language tasks. Conversely, HuggingGPT can break this boundary and accommodate various modalities and scenarios from the task list.
> > > >
> > > > + Lastly, regarding references about task decomposition. Thanks for your suggestions, we are willing to add references on this topic in our final version.
> > > >
> > > >
> > > > | Request | GPT-3.5 | HuggingGPT | HuggingGPT w/o task planning |
> > > > |---------|---------|------------|-----------------------------|
> > > > | Kindly provide information concerning the pathogenicity-associated proteins of the Omicron variant of the SARS-CoV-2 virus. | As of my last knowledge update in September 2021, I don't have specific information about the Omicron variant of SARS-CoV-2, as it emerged after that date. However, I can provide you with some general information about SARS-CoV-2 and its variants up to that point. SARS-CoV-2 is the virus responsible for the COVID-19 pandemic. It primarily infects human respiratory cells and causes a range of symptoms, from mild respiratory issues to severe pneumonia and in some cases, death. The virus uses various proteins to enter human cells, replicate, and evade the immune response. | In response to your request, I will first generate a pathogenicity report on Omicron variants of SARS-COV-2 for you. Then I will list the proteins involved in the report. Initially, I employed 'microsoft/BioGPT' to craft a report concerning the pathogenesis of the Omicron variant of the SARS-CoV-2 virus, utilizing the introductory statement, "Extensive research has elucidated the pathogenesis of the Omicron variant of the SARS-CoV-2 virus.". The generated report is shown below: Extensive research has elucidated the pathogenesis of the Omicron variant of the SARS-CoV-2 virus. Like other SARS-CoV-2 variants, Omicron uses the spike protein to bind to the angiotensin-converting enzyme 2 (ACE2) receptor on human cells, primarily those in the respiratory system. The high number of mutations in the spike protein raised concerns about increased affinity for the ACE2 receptor, possibly enhancing the virus's ability to enter and infect human cells. Subsequently, we employed MilosKosRad/BioNER to extract proteins from the report, yielding the following results: 1. Spike protein 2. Angiotensin-converting enzyme 2 (ACE2) receptor These are the primary proteins associated with Omicron entry into human cells. | Failed |
> > > > |Please provide a description of the image "examples/test.jpg," and then assist me in creating an anime-style illustration along with accompanying dialogue. | I apologize for any misunderstanding, but as a text-based AI, I don't have the capability to view images, including "examples/test.jpg," or any other external content. I can only work with the text description and information you provide. If you describe the scene, characters, and elements present in the image, I can certainly help you come up with a description, create a dialogue, or even offer advice on how to illustrate it in an anime-style manner. Just provide me with the details, and I'll be happy to assist!| For your request, I organized three steps: converting image to text, transforming text to image, and generating text. Initially, I employed "nlpconnect/vit-gpt2-image-captioning" for description creation. The resulting description is: "A group of boys playing basketball". Subsequently, we utilized the anime style image generation model to create an image from the given description. You can find this image at "images/fsuq.jpg". To conclude, we harnessed the ChatGPT model to craft a pertinent dialogue: "Alex: "Hey Jake, pass the ball!"Jake: "You got it!"Liam: "Come on, Alex, shoot! I've got the rebound if you miss."Alex: "Alright, here goes!"Ryan: "Nice one, man! That's three points."|Failed|

---

> > > > > ### Comment · Reviewer_hS4Z · 2023-08-21
> > > > > **A quantitative evaluation outweighs thousands of words.**
> > > > >
> > > > > The author presented a large body of text trying to motivate not doing the simple ablation study. I am not convinced at all and become suspecting that a simple one-pass reasoning by LLM can not be easily outperformed, unless there is a quantitative measurement to rule out my argument.
> > > > >
> > > > > Presenting this baseline will not:
> > > > > - break the integrity of the proposed approach. This is an ablation study, isn't it?
> > > > > - downweight the importance of planning in solving complex tasks. There are plenty of works in planning have already shown this. But how can I know if you are doing in the right way? Let me say this: your proposed approach could be orthogonal to the importance of planning in solving the test cases, unless you prove that they are correlated.
> > > > >
> > > > > I think the two examples are heading the right direction. Please scale it up. Otherwise, I vote for rejection.

---

> > > > > > ### Author Response · Authors · 2023-08-21
> > > > > > **Response to Comments**
> > > > > >
> > > > > > **Regarding the Effectiveness in your limitations**:
> > > > > >
> > > > > > - If you mention the effectiveness of HuggingGPT, our experiments (Table 3-6 and Figure 1-2 and supplemental material) can show that HuggingGPT is able to address AI tasks with multiple modalities and domains, and thus demonstrate its effectiveness.
> > > > > >
> > > > > > - If you mention the effectiveness of task planning, let us first review our motivation, which is how to extend the task scope of LLMs to multiple modalities (e.g., text, image, audio, and video) and domains. So the critical point is to verify whether our proposed planning can fulfill this point. To implement this, we provide a task list in our planning and thus require it to understand each task first and then parse user requests into task sequences, including task type, arguments, and dependency. The results in Table 3-6 show that when using ChatGPT or GPT-4, the designed task planning can achieve 50+% accuracy in understanding user requests. So, this point demonstrates that our proposed planning can reach our motivation. However, we do not argue that our planning is optimal but just present a reasonable strategy to extend the task scope of LLMs, which is the focus of this work. We also have given a discussion about the difference between our planning and other alternative planning (e.g., iterative planning in AutoGPT) in the supplemental material, and they have some trade-offs in efficiency, stability, and capability. We will leave the design of more planning strategies as future work.
> > > > > >
> > > > > > **Regarding the Ablation Study in your response:**
> > > > > >
> > > > > > - It is confusing to just compare with ChatGPT on language-only tasks. Since ChatGPT is also a model (tool) in our model selection, it has two cases in the ablation study: 1) ChatGPT is selected as the expert model; 2) other expert models are selected. So, the design of our framework guarantees the lower bound is ChatGPT.
> > > > > >
> > > > > > - For other scenarios (please see our provided results), it only has two results: 1) HuggingGPT without planning will directly fail; 2) when switching to multi-modality tasks, ChatGPT will fail.

---

> > > > > > > ### Comment · Reviewer_hS4Z · 2023-08-21
> > > > > > > **Up to the Chairs now**
> > > > > > >
> > > > > > > Some of my arguments got throw back at me. I think my question is a very simple one:
> > > > > > > "One very simple and easy-to-come-with experiment is comparing against solving some sequential/graph language-only tasks without any planning. "
> > > > > > >
> > > > > > > Consider this as a naive baseline to beat. Otherwise, suppose someone issues an argument that, *HuggingGPT performs worse than relying on a LLM to do implicit reasoning when dealing with complex language tasks*. How to prove it wrong? with only 2 examples?
> > > > > > >
> > > > > > > Again, as I should always mention, the proposed idea is an interesting one. It definitely has value in the thought.

---

> > > > > > > > ### Author Response · Authors · 2023-08-21
> > > > > > > > **Response to Comments**
> > > > > > > >
> > > > > > > > To address your concerns, we selected 46 language-only samples from our evaluation dataset (refer to our response to FNMr about modal distributions) and produced responses via HuggingGPT and ChatGPT. Borrowing from alpacaeval, we performed a comparative evaluation of ChatGPT and HuggingGPT responses. We used the human evaluator and the GPT4 evaluator, respectively, to select the winner from ChatGPT and HuggingGPT, and then calculated the win rate. For details on the human evaluations, please see our response to FNMr. For the GPT4 evaluation, we set up the following prompts:
> > > > > > > >
> > > > > > > >
> > > > > > > >
> > > > > > > > >Assuming you are a well-trained evaluator, you are to determine which of A's or B's responses resolves the request #REQUEST#. Please adhere to the following guidelines: 1. The criterion for determining if a request is resolved is whether the question in the request has been fully answered. It must not be partially answered or outright declined. 2.The response must adhere to basic logic and exhibit consistency. 3.If both A and B resolve the request, select the most plausible one based on your judgment. 4.If neither response resolves the request, indicate "None." # REQUEST #
> > > > > > > > {request}
> > > > > > > > >#Response from A#
> > > > > > > > >{response_a}
> > > > > > > > >#Response from B#
> > > > > > > > >{response_b}
> > > > > > > > >Your Answer: (A/B/AB/None)"
> > > > > > > >
> > > > > > > >
> > > > > > > >
> > > > > > > > Additionally, we anonymized HuggingGPT and ChatGPT to avoid the influence of human and GPT4 priors on the evaluations.
> > > > > > > >
> > > > > > > >
> > > > > > > >
> > > > > > > > In the case of human and GPT4 evaluations, we report the win rate as:
> > > > > > > >
> > > > > > > >
> > > > > > > >
> > > > > > > > |Model|Rate|
> > > > > > > > |--|--|
> > > > > > > > |HuggingGPT win|56.5%|
> > > > > > > > |HuggingGPT win+tie|84.8%|
> > > > > > > >
> > > > > > > >
> > > > > > > >
> > > > > > > > |Model|Rate|
> > > > > > > > |--|--|
> > > > > > > > |HuggingGPT win|60.9%|
> > > > > > > > |HuggingGPT win+tie|87.0%|
> > > > > > > >
> > > > > > > >
> > > > > > > >
> > > > > > > > Based on the results, we can find that HuggingGPT can achieve better results than ChatGPT in the win (plus tie) rate. Since ChatGPT has been considered as one of the expert models in HuggingGPT, sometimes, HuggingGPT will also choose ChatGPT to address some language tasks.
> > > > > > > >
> > > > > > > >
> > > > > > > >
> > > > > > > > Moreover, the core contribution of HuggingGPT is to establish the connection between LLMs and other models. We can also enhance the capability of HuggingGPT in solving more complex language tasks by adding more powerful language agents (e.g., Reflection, etc) to the model library.

---

### Author Rebuttal · Authors · 2023-08-10

# To All Reviewers


We sincerely thank each reviewer for providing constructive comments for our paper, which are very helpful to improve our paper. Below are our responses to some general issues:

***Q1: Model Selection***

Thanks for the comments of each reviewer. Here, we will provide more details and descriptions to better explain the design of model selection and its correlation with task planning. We think it is necessary to keep the model selection as an individual step and give these reasons:

1. **Motivation**: First, task planning helps us to analyze user request and decompose it into a task sequence, while model selection is to answer which model is suitable for this task. Therefore, the target of each stage is different, and we prefer to keep two steps respectively and thus encourage to design different strategies for each.

2. **Efficiency**: Sometimes, the number of models from ML communities could be massive. Since we need to use model descriptions to call the most suitable model, selecting models will bring more token consumption in prompts. As a result, combining task planning and model selection will cost more tokens within one query, and separating these two steps will be more friendly for LLM computations.

3. **Flexibility and Extensibility**: Furthermore, keeping task planning and model selection as two parts can provide better flexibility and extensibility: 1) the model library is usually dynamic since developers will continually add the latest models into the community. Therefore, we expect that the model selection is independent of task planning so that it is easy for us to update the model library; 2) some models could include more detailed granularity (e.g., specific domains). These settings necessitate the development of more flexible selection strategies.

We will add these descriptions to the final version to better explain the design of our framework.

***Q2: Experiments & Evaluation***

Thanks for the suggestions provided by each reviewer. We have added these results to better explain the settings in our framework.

+ Table 1 and Figure 1 in Rebuttal PDF show the effects of variance and number of demonstrations.
+ Table 2 in Rebuttal PDF demonstrates the evaluation of our framework by human experts.
+ Table 3 in Rebuttal PDF presents some cases illustrating the model selection.

Specifically, for human evaluation, we collected 130 diversity requests to measure the planning capability and final generation of LLMs in our framework. Here, we design passing rate, rationality and success rate to evaluate different stages:

- Passing rate: Check LLMs whether can understand instructions and execute in the current stage.
- Rationality: Check whether the generated task sequence / selected tools are rational when compared with user requests.
- Success rate: Check the final results whether satisfy the user request.

We invite three human experts to annotate the provided data based on our designed metrics, and then calculate the average value as the final score.

Please refer to the attached Rebuttal PDF for the detailed results.

| **Demo Variance** | **LLM** | **Single Task**    |  | **Sequential Task**   |  | **Graph Task** |  |
|:---------------------------------------|:---------|:-------------|:---------|:-----------------|:--------:|-----------|-----|
|                                       |         | **Acc** ↑ | **F1** ↑  | **ED (%)** ↓ | **F1** ↑  | **F1** ↑ |
| 2                                     | GPT-3.5  | 43.31      | 48.29     | 71.27        | 32.15     | 43.42         |
| 2                                     | GPT-4    | 65.59      | 67.08     | 47.17        | 55.13     | 53.96         |
| 6                                     | GPT-3.5  | 51.31      | 51.81     | 60.81        | 43.19     | 58.51         |
| 6                                     | GPT-4    | 66.83      | 68.14     | 42.20        | 58.18     | 64.34         |
| 10                                    | GPT-3.5  | 52.83      | 53.70     | 56.52        | 47.03     | 64.24         |
| 10                                    | GPT-4    | 67.52      | 71.05     | 39.32        | 60.80     | 66.90         |

*Table 1. Evaluation of task planning in terms of the diversity of demonstrations. We denote demonstration variance as the number of different task types involved
in the demonstrations. (Same as Table 1 in Rebuttal PDF.)*


| **LLM**      | **Task Planning** |              | **Model Selection** |        | **Response Generation**  |
|--------------|-------------------|--------------|---------------------|--------|---------------------------|
|              | **Passing Rate** ↑  | **Rationality** ↑  | **Passing Rate** ↑  | **Rationality** ↑  | **Success Rate**↑     |
| Alpaca-13b   | 51.04      | 40.26             | -           | -                | 6.92                      |
| Vicuna-13b   | 79.41      | 66.15             | -           | -                | 15.64                     |
| GPT-3.5      | 99.42      | 91.03             | 93.89       | 84.29            | 63.08                     |

*Table 2. Human Evaluation. We design passing rate, rationality and success rate as the subjective metrics and invite human experts to evaluate them. (Same as Table 2 in Rebuttal PDF.)*

---

### Comment · Area_Chair_LDRN · 2023-08-18
**Discussion period**

Dear reviewers,

Now we are in the discussion period. Please read the authors' rebuttal, others' reviews, and raise questions if need, and/or acknowledge the rebuttal has read, etc. Thank you for your time.

AC

---

### Decision · Program_Chairs · 2023-09-21

**Decision:**

Accept (poster)

**Comment:**

Summary of reviews:

The proposed framework, HuggingGPT, is a novel approach to using large language models (LLMs) to solve complex tasks. The framework consists of four steps: task planning, model selection, task execution, response generation.

Strengths:

- It is able to solve a wide variety of complex tasks, including tasks that require multiple modalities (e.g., text, images, and audio).
- It is able to leverage the strengths of LLMs, such as their ability to learn from large amounts of data and generate text outputs.
- It is able to select the appropriate models to execute each subtask, which can improve the performance of the framework.

Weaknesses:

- It is dependent on the availability of LLMs. This could be a limiting factor in some cases, as LLMs can be expensive to train and maintain.
- It is not yet clear how the framework would perform in real-world settings.

Overall, the proposed framework is a promising approach to using LLMs to solve complex tasks. More research is needed to evaluate the framework in real-world settings and to address its weaknesses. I thank the authors for engaging with reviewers extensively to address their questions and provide additional results. Hence, I recommend this paper for acceptance.